



# Spatio-temporal melt and basal channel evolution on Pine Island Glacier ice shelf from CryoSat-2.

Katie Lowery[1,2], Pierre Dutrieux[1], Paul R. Holland[1], Anna E. Hogg[2], Noel Gourmelen[3], and Benjamin J. Wallis[2]

[1]British Antarctic Survey, Cambridge
[2]School of Earth and Environment, University of Leeds
[3]School of Geosciences, University of Edinburgh

**Correspondence:** Katie Lowery (katlow20@bas.ac.uk)

**Abstract.**

Ice shelves buttress the grounded ice sheet, restraining its flow into the ocean. Mass loss from these ice shelves occurs primarily through ocean-induced basal melting, with the highest melt rates occurring in regions that host basal channels — elongated, kilometre-wide zones of relatively thin ice. While some models suggest that basal channels could mitigate overall

ice shelf melt rates, channels have also been linked to basal and surface crevassing, leaving their cumulative impact on ice-shelf stability uncertain. Due to their relatively small spatial scale and the limitations of previous satellite datasets, our understanding of how channelised melting evolves over time remains limited. In this study, we present a novel approach that uses CryoSat-2 radar altimetry data to calculate ice shelf basal melt rates, demonstrated here as a case study over Pine Island Glacier (PIG) ice shelf. Our method generates monthly Digital Elevation Models (DEMs) and melt maps with a 250 m spatial resolution.

The data show that near the grounding line, basal melting preferentially melts a channel's western flank 50% more than its eastern flank. Additionally, we find that the main channelised geometries on PIG are inherited upstream of the grounding line and play a role in forming ice shelf pinning points. These observations highlight the importance of channels under ice shelves, emphasising the need to investigate them further and consider their impacts on observations and models that do not resolve them.

## 1 Introduction

Ice shelves, the floating extensions of ice sheets, surround the majority of the Antarctic coastline (Rignot et al., 2013) and act to buttress the grounded ice (Dupont and Alley, 2005; Goldberg et al., 2009; Gudmundsson, 2013). Between 1997 and 2021, Antarctic ice shelves lost a mass of 7,500 Gt (Davison et al., 2023), predominantly driven by basal melting induced by ocean forcing (Rignot et al., 2013; Davison et al., 2023). Notably, 90% of Antarctic mass loss since 1992 was concentrated in West

Antarctica (WA) (Shepherd, 2018), where the rate of mass loss increased by 70% between 1994 and 2012 (Paolo et al., 2022). By 2017 WA had already contributed between 5.7 mm and 6.9 mm to global sea level rise (Shepherd et al., 2019; Rignot et al., 2019). The mass loss trend in WA, and in particular the Amundsen Sea region, is thought to be driven by the intrusion of relatively warm Circumpolar Deep Water onto the continental shelf (Dutrieux et al., 2014a; Jacobs et al., 1996; Shepherd





et al., 2004). Further retreat, and potential collapse, of many ice shelves in the Amundsen Sea has been predicted (Joughin
et al., 2010; Naughten et al., 2023; Joughin et al., 2014; Favier et al., 2014; Bett et al., 2024; De Rydt and Naughten, 2024).

Pine Island Glacier (PIG) in the Amundsen Sea has shown almost continuous acceleration (Joughin et al., 2003; Rignot et al.,
2008; Mouginot et al., 2014) and thinning (Wingham et al., 2009) since the 1990s. By 2009, PIG's ice shelf flowed at 4 km/yr
following a 75% acceleration since the 1970s (Mouginot et al., 2014). This acceleration was accompanied by a 50% increase
in ice discharge at the grounding line (Medley et al., 2014). PIG's central grounding line also retreated by 31km (Rignot et al.,
2014) between 1996 and 2011. During this retreat a pinning point, a localised area of grounded ice, formed in the centre of the
ice shelf. Pinning points usually form on areas of anomalously high seabed topography. Pinning points significantly contribute
to the buttressing capacity of an ice shelf (Favier et al., 2016; Schlegel et al., 2018) and have been shown to dictate the future
of Thwaites Glacier (Bett et al., 2024). The pinning point in the centre of PIG ungrounded in 2011 and since then its thick
column of ice has advected downstream, ephemerally regrounding (Rignot et al., 2014; Joughin et al., 2016). Following this,
major calving events between 2017 and 2020 resulted in the ice front retreating a further 19 km and the ice shelf accelerating
by 12% (Joughin et al., 2021).

On ice shelves, basal channels are elongated areas of relatively thin ice typically a few kilometres in width and spanning up to
the whole length of the ice shelf. They have been observed on both Greenland and Antarctic ice shelves (e.g. Gourmelen et al.,
2017; Dutrieux et al., 2013; Vaughan et al., 2012; Millgate et al., 2013; Drews et al., 2017; Alley et al., 2016; Motyka et al.,
2011), including PIG ice shelf which has an abundance of basal channels (Dutrieux et al., 2013; Shean et al., 2019; Vaughan
et al., 2012; Stanton et al., 2013). Channels modulate basal melt rates and, near the grounding line, focus the highest melt
rates of an ice shelf within them (Dutrieux et al., 2013). Inside these channels, melt-induced buoyant freshwater plumes rise
along the ice shelf base, entraining relatively warm water that enhances melting (Millgate et al., 2013; Gladish et al., 2012).
However, further downstream, basal melt rates are observed to be higher on basal keels than within the channels (Dutrieux
et al., 2013; Gladish et al., 2012). Channels can form from a variety of processes indicated by their inception location. Some
form downstream of subglacial outflow locations (Le Brocq et al., 2013; Gourmelen et al., 2017; Marsh et al., 2016). Others
form downstream of eskers on the seabed of grounded ice (Drews et al., 2017). Finally, a third type may be carved by buoyant
ocean plumes resulting from instabilities of the coupled ice/ocean system (Alley et al., 2016; Sergienko, 2013). Gladish et al.
(2012) and Millgate et al. (2013) showed that channels can reduce ice shelf area-integrated melt rates and also distribute
melting more widely across an ice shelf, thus increasing the overall stability of an ice shelf. However, others have suggested
channels can reduce the structural integrity of an ice shelf through increased surface and basal crevassing (Alley et al., 2016;
Vaughan et al., 2012). While ocean-driven basal melting stands as the primary contributor to mass loss in Antarctica, precise
measurements of basal melt pose ongoing challenges. Various remote sensing techniques have been employed to derive melt
rates (Shean et al., 2019; Zinck et al., 2023; Dutrieux et al., 2013; Paolo et al., 2022), alongside methods using sub-ice shelf
oceanographic observations (Davis et al., 2023; Dutrieux et al., 2014a; Schmidt et al., 2023), ocean observations in front of ice
shelves (Jenkins et al., 2018) and direct in-situ observations of melting using Automatic phase-sensitive Radio Echo Sounding
(ApRES; Nicholls et al., 2015; Davis et al., 2018; Vaňková et al., 2021). Remote sensing techniques have given us a broad
understanding of Antarctic-wide phenomena (Rignot et al., 2013; Adusumilli et al., 2020; Smith et al., 2020), but they have





struggled to deliver melt rate fields with high spatial and temporal resolution. In-situ ApRES observations, on the other hand, have given us measurements with very high temporal resolution and accuracy but are constrained to a few point locations.

Remote sensing techniques are ever-improving, but there remains a trade-off between spatial and temporal resolution. Some ice shelf basal melt studies prioritise achieving high spatial resolution (<1 km) at the expense of temporal resolution (Dutrieux et al., 2013; Adusumilli et al., 2020; Zinck et al., 2023; Gourmelen et al., 2017), while others sacrifice spatial resolution to maintain adequate temporal resolution (Paolo et al., 2022). Due to these constraints, time-varying derivations of melting in basal channels are limited, leading to a lack of understanding of how channelised melting evolves and impacts ice-shelf stability. In this study, we introduce a novel methodology aimed at optimising both the spatial and temporal resolution of basal melt rates derived from surface elevation altimetry data. We apply this methodology to high-resolution CryoSat-2 Synthetic Aperture Radar Interferometer (SARIn) swath processed radar surface elevation measurements. This enables us to derive monthly Digital Elevation Models (DEMs) and basal melt maps at a spatial resolution of 250m. This study is completed as a case study over PIG; however, we note that this method can be applied to any surface elevation altimetry data where accurate velocity and divergence measurements exist. Leveraging this dataset, we explore the formation and growth of ice shelf basal channels, uncovering new insights into their evolution on PIG and their potential importance for the interaction and formation of ice shelf pinning points.

## 2 Data and Methods

### 2.1 CryoSat-2 Surface Elevation Data

We generate DEMs using CryoSat-2 (CS2) SARIn swath-processed radar altimetry data (Gourmelen et al., 2018). Over the Pine Island embayment area shown in Figure 1, there are approximately 180 million elevation measurements between 2011 and 2023. These measurements are used to create monthly DEMs with a spatial resolution of 250 m. To achieve full coverage at this spatial resolution, each DEM requires a year's worth of data. We therefore produce a product with an annual resolution at monthly posting, as described below.

Before creating our DEMs many corrections are first applied to the CS2 observed ice shelf surface elevation. Each CS2 elevation measurement has been corrected for tides, the inverse barometer effect (IBE) and firn air content. Tidal corrections have been applied from CATS model outputs (Howard et al., 2019) and IBE corrections have been taken from the CNES SSALLTO system provided by Meteo France. Firn air content has been taken from the Institute for Marine and Atmospheric research Utrecht (IMAU) firn densification model forced by RACMO model outputs (Veldhuijsen et al., 2023).

The 12 months of data that is required to created a single DEM is centred around the target date the DEM represents. The target date is the 1st of the month and all elevation data within 6 months of this date are gathered and advected to the location they would have been on the target date using ice velocity data. For example, if we were creating a DEM on July 1st, all observations acquired between January 1st and June 30th of the same year would be advected forward to their location on July 1st and all observations between July 2nd and December 31st would be advected back to their location on July 1st. This process will be referred to as 'time-centering' from here onwards. Specifically, if a given elevation measurement, $h$, is recorded





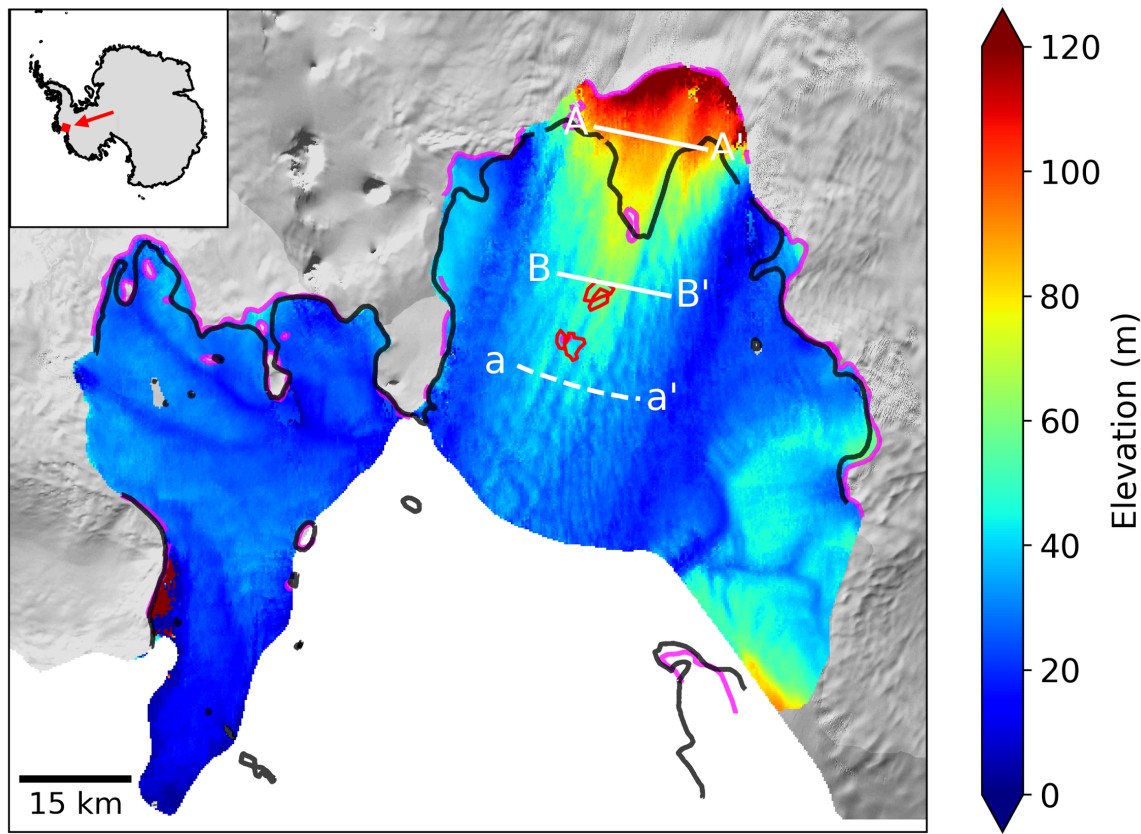

**Figure 1.** Digital Elevation Model of Pine Island Glacier in 2017 derived from CryoSat-2. The 1996 and 2011 MEaSUREs grounding lines (Rignot et al., 2016) are shown in black and magenta, respectively. The areas of ephemeral re-grounding in 2017 as observed from Sentinel-1 using the DROT method (described in Section 2.3) are shown in red. The white transects correspond to sections used in Figure 7. Background: greyscale Landsat Image Mosaic of Antarctica (LIMA) 15m.

in month $m_0$ with a location of $(x_0, y_0)$ and the DEM being created is centred on month $m_n$, its time-centred location is given by:

$$(x_n, y_n) = (x_0 + \sum_{i=0}^{n-1} u_i dt, y_0 + \sum_{i=0}^{n-1} v_i dt),$$ (1)

where $u_i$ and $v_i$ are the velocities in the $x$ and $y$ directions evaluated at the $(x, y)$ location of $h$ at time step $i$ and $dt$ is the time step duration. Once this advection procedure has been applied to all measurements within 6 months of the target date, the resultant data points are gridded into 250 m x 250 m bins and that bin takes the median value of all its members. Advection is calculated using ITS_LIVE annual velocity mosaics (Figure 2b) and with a monthly time step. With CS2 surface elevation data spanning January 2011 to December 2023, our monthly DEMs start from July 2011 and extend to July 2023.



We have chosen to implement this time-centering method because it more accurately represents the surface of the ice shelf when combining a year of data than by directly binning the observations. Temporal averaging is required to obtain 250 m horizontal resolution but, over a year of coverage, a parcel of ice could have crossed 16 of the 250 m grid boxes (assuming a velocity of 4 km/yr). By simply binning these observations, ice shelf surface geometries will be smoothed over a few kilometres, removing our ability to observe changes within channels. By adopting the time-centering process, we reduce the aliasing of

these features (Figure A1).

To create the DEM, we advect elevation measurements to their predicted positions at the target date, ignoring any thickness changes during the advection process. Although this assumption introduces some uncertainty, its impact on the final DEM is minimised because the advection is performed both forward and backward in time. Specifically, a measurement advected forward by six months (and therefore experiencing six months less melt) is balanced by a measurement advected backward by

six months (experiencing six months more melt). By taking the median value within each bin, these opposing effects largely cancel each other out, resulting in a robust representation of elevation.

## 2.2    Basal Melt Calculations

Once monthly DEMs are generated, we utilise them (e.g. Figure 2a) to calculate Lagrangian basal melt rates (Figure 2d) by employing a similar method to those detailed previously (Shean et al., 2019; Dutrieux et al., 2013; Zinck et al., 2023; Moholdt

et al., 2014; Gourmelen et al., 2017). The Lagrangian framework measures thickness change of a parcel of ice as it moves, thereby avoiding Eulerian problems associated with the advection of ice topographic features through the grid. Assuming floatation of the ice, basal melt rates, $m$, are given by:

$$m = -\frac{DH}{Dt} - H(\nabla \cdot \mathbf{u}) + s, \tag{2}$$

where

$$H = h\left(\frac{\rho_w}{\rho_w - \rho_i}\right) \tag{3}$$

is the ice thickness, $\frac{DH}{Dt}$ is the Lagrangian thickness change, $\mathbf{u}$ is the horizontal ice velocity vector, $H(\nabla \cdot \mathbf{u})$ is the thickness change resulting from flow divergence and $s$ is the Surface Mass Balance (SMB), positive for accumulation. SMB has been taken from RACMO model outputs (Noël et al., 2018). By assuming hydrostatic equilibrium and constant ocean and ice densities, $\rho_w = 1028$ kg/m$^3$ and $\rho_i = 917$ kg/m$^3$ respectively, ice elevation, $h$, is converted into ice thickness, $H$, following

equation (3).

For each month, we conduct the same advection process as described in Section 2.1 to predict the location of an ice parcel after a year. Subsequently, the Lagrangian thickness change is determined by comparing the ice thickness at the original location with the thickness at its location a year later. This process yields a Lagrangian change in thickness $\left(\frac{DH}{Dt}\right)$ map that is temporally centered between the two DEMs, at the 'mid-time' stamp half a year after the start date. The Lagrangian thickness

change is allocated to its original location in space, following the 'initial pixel' method outlined by Shean et al. (2019). The





**Figure 2.** Example basal melt calculation components. (a) Digital Elevation Model derived from CryoSat-2, (b) velocity, coloured by velocity magnitude and overlaid with velocity vectors, (c) velocity divergence, and (d) basal melt rates. All components shown are in 2015. The black rectangle in (a) is for Figures 5 and 6. Basemap and grounding lines as in Figure 1.

change in ice thickness as a result of velocity divergence ($H(\nabla \cdot \mathbf{u})$, Figure 2 c) is considered based on the original location of the ice parcel and the mid-time stamp and is again calculated using the annual ITS_LIVE velocity mosaics (Figure 2b).

Our melt maps at monthly postings remain relatively noisy at the channel scale. To further reduce the noise and obtain a coherent channelised melt signal, we further averaged 12 months of melt maps. To complete this averaging, we have advected 135 the 12 months of melt data to their location at the middle time step using the same advection process described in Section 2.1. The advected data is then re-binned, and the median melt value in the bin is taken. This averaging has been completed on a



rolling 12 months basis and therefore we retain monthly posting between July 2012 and December 2021. Therefore, Cryosat-2 elevation data spanning 3 years has contributed to each monthly posted melt map.

We use annual velocity datasets throughout our method, which introduces discrete jumps at annual boundaries. During the advection process (both for time-centering and melt advection) the velocity dataset corresponding to the year of each time step is used. Similarly, the divergence field for the melt calculation is taken from the year of the target month. This approach avoids averaging the divergence field, which would smooth the channel-scale features we aim to preserve. This choice ensures consistency throughout the method by aligning the treatment of velocity and divergence fields.

Obtaining basal melt rates with this approach is based on several key assumptions. Specifically, it assumes that the ice is fully floating and in hydrostatic equilibrium, that the ice velocity is vertically uniform (and therefore the surface velocity is a good representation of the ice velocity at the base), and that the ice is a viscous continuum (does not fracture in response to divergence) (Dutrieux et al., 2013; Zinck et al., 2023). Bridging stresses within the ice make deriving changes within small channels through changes in surface elevation challenging (Dutrieux et al., 2013). It is important to acknowledge these limitations inherent to methods using remote sensing techniques to derive ice shelf basal melt rates, particularly when attempting to infer changes to features of similar scales to a few ice thicknesses. In-situ surveys demonstrate the complexity of the basal geometry associated with smoother surface expressions (Dow et al., 2024; Dutrieux et al., 2014b; Drews, 2015).

## 2.3 Sentinel-1 Differential Range Offset Tracking

To investigate the relationship between channels and pinning points, we derive the locations of pinning point groundings using the differential range offset tracking (DROT) method applied to Sentinel-1 synthetic aperture radar (SAR) data. The DROT technique makes use of SAR sensors' off-nadir viewing geometry to detect floating ice, because vertical tidal displacements are projected into the satellite's line-of-sight and appear as an anomalous horizontal ground-range displacement in intensity feature tracking results (Joughin et al., 2016; Friedl et al., 2020; Marsh et al., 2013). By differencing adjacent feature tracking results, or subtracting a reference steady-state ground-range velocity field, the anomalous ground-range displacement caused by vertical tidal displacement is isolated, in a process analogous to differential interferometric SAR (DInSAR). From this tidal displacement result, the grounding line and any pinning points can be delineated. Because DROT uses intensity feature tracking, rather than InSAR, to measure ice motion, it does not require InSAR coherence and is suitable for investigating pinning points on fast flowing ice shelves, such as PIG, where coherence is not routinely maintained in the 6-day repeat period of Sentinel-1.

In this study, we use 6 and 12 day repeat Sentinel-1 intensity feature tracking pairs over the PIG shelf processed using the GAMMA remote sensing software package (Strozzi et al., 2002). Tracking results are post-processed using a moving mean filter over a 1x1 km window, where values $30\%$ greater than the mean are rejected and isolated pixels which represent poor tracking results are also removed according to a locally determined threshold (Lemos et al., 2018; Selley et al., 2021). Outputs are posted at a resolution of 100x100 m in Antarctic Polar Stereographic projection (EPSG:3031). To produce observation of ephemeral grounding and pinning points we select equitation pairs with differential tide heights greater than 0.5 m and remove the steady-state ground range velocity field by subtracting the median ground range velocity from a 180 day window centred





on the measurement date to calculate the tidal motion anomaly. This anomaly is manually delineated with GIS software to determine the location of any grounded portions of the ice shelf for the selected acquisition pair.

## 3 Results

### 3.1 Channel and pinning point history

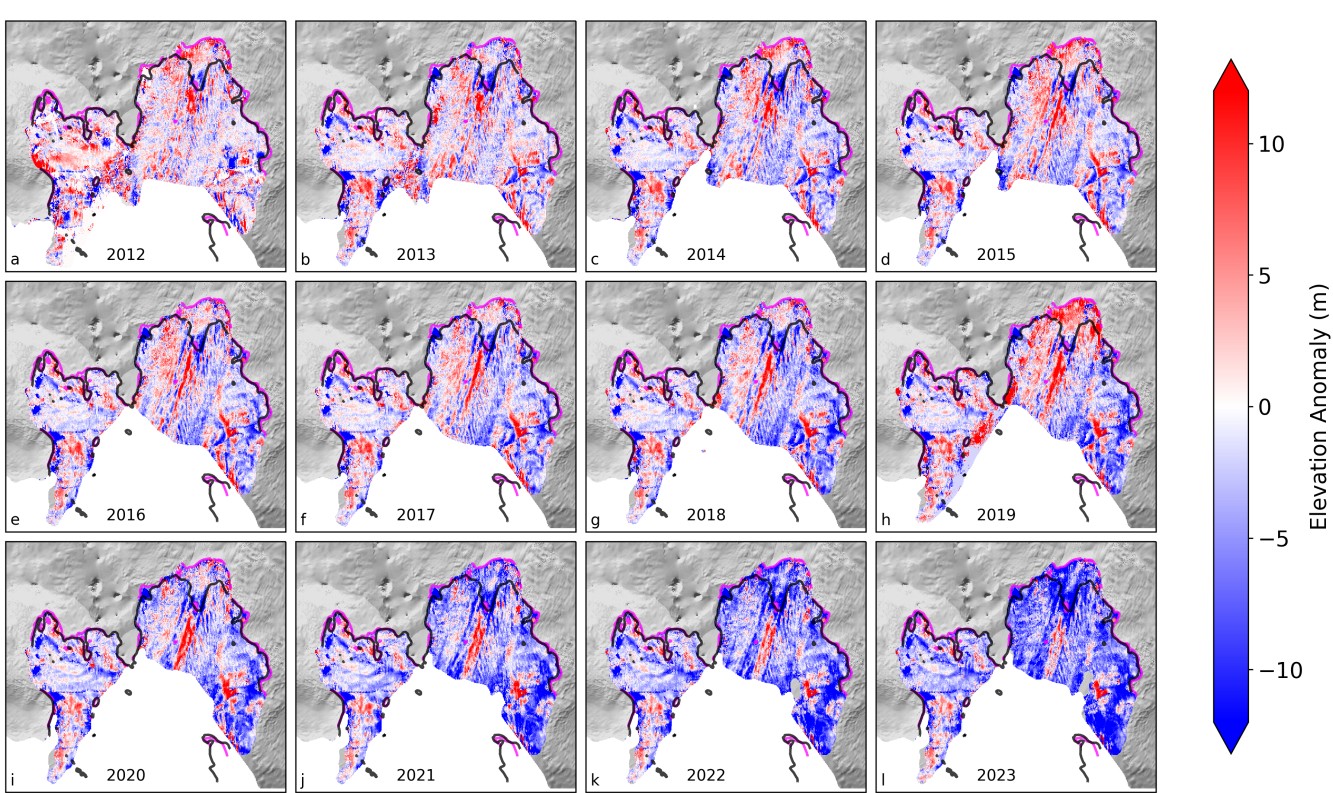

**Figure 3.** (a)-(l) show the annual elevation anomalies with respect to the first DEM in June 2011 for 2012-2023, respectively. Basemap and grounding lines as in Figure 1.

Figure 3 shows the annual elevation anomaly with respect to the first DEM in our time series (July 2011). Over the observed period, the ice shelf gradually thinned, and the calving front retreated. On average the ice shelf thinned by 60 m between 2011 and 2023 (Figure 3l), the equivalent to a mass loss of 259 Gt.

In addition to overall thinning, geometric features such as basal channels have evolved, imprinting small-scale features on top of the overall thinning rates (Figure 3). The pinning point in the centre of the ice shelf began to unground in 2011 (Joughin et al., 2016), resulting in a thick column of ice being advected downstream. This is visible as the anomalously thick (red) region




in the centre of Figure 3a. Over time, the entire downstream section of the ice shelf thickened as the thicker ice was transported further downstream.

Before 2011, a large basal channel extended downstream of the pinning point, as shown in Figure 4a (beyond 8km) and in the centre of Figure 4d. We hypothesise that this channel formed due to interactions between the ice and bed at the pinning
point, which imprinted on the ice geometry, and was further enhanced by concentrated melting within the channel. After the pinning point ungrounded, this basal channel was gradually advected downstream and replaced by thicker ice. Simultaneously, a new arrangement of basal channels began forming further upstream (Figures 3 and 4).

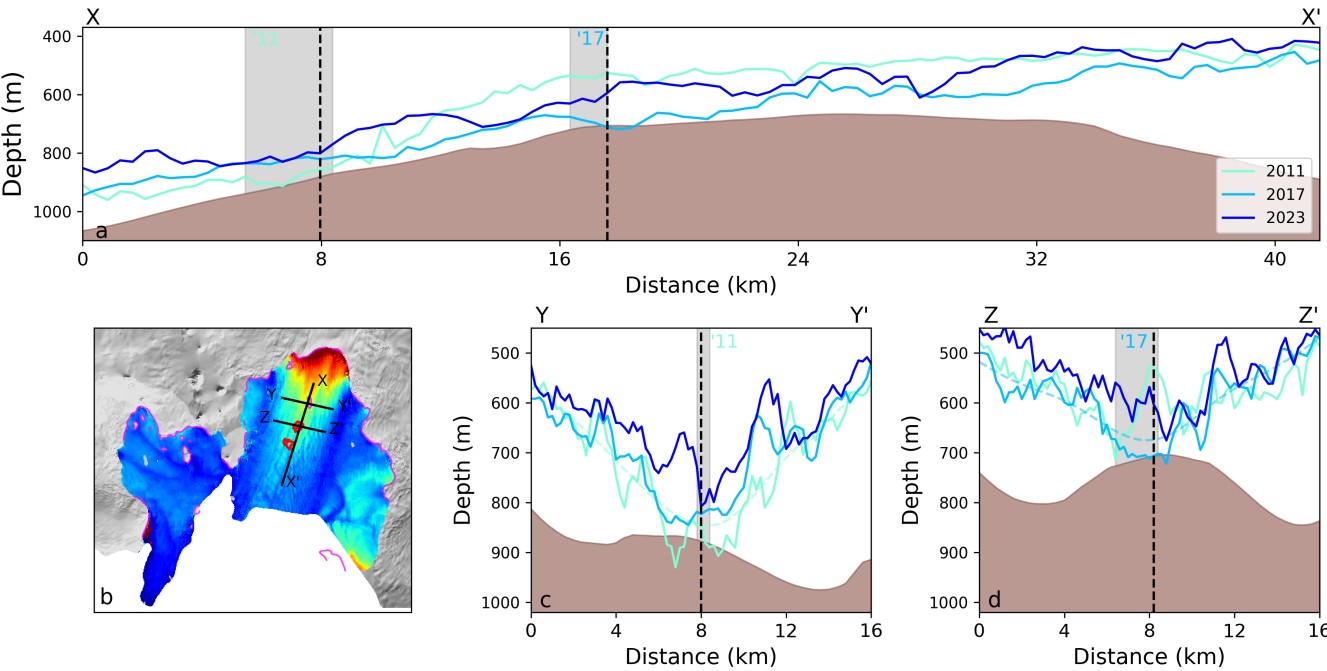

**Figure 4.** Depth of the ice shelf base in 2011, 2017 and 2021 across three different transects. (b) shows the three transects used in the other subplots. (a) shows the depth of the base between X and X', (c) shows the depth of the base between Y and Y', and (d) shows the depth of the base between Z and Z'. The red lines in (b) show the areas where Sentinel-1 DROT data indicates grounded ice in 2017. The vertical grey shaded areas in (a),(c) and (d) show grounded areas in 2011 and 2017 as observed by the MEaSUREs 2011 grounding line and DROT 2017. The black dashed vertical lines mark the points of intersection between transects. The brown shaded areas on (a), (c), and (d) are the seabed from BedMachine merged with observations from (Dutrieux et al., 2014b).

To investigate the relationship between basal channel distribution and grounding, we will focus on the change in the ice base depth over our observational period across 3 transects (Figure 4). The depth of the ice shelf base has been calculated by
inverting the surface elevation and assuming floatation. Across all 3 transects, the depth of the ice shelf in 2011 (light blue), 2017 (blue) and 2023 (dark blue) have been plotted. These years represent when the pinning point was starting to unground (2011), when the pinning point ephemerally regrounded in the centre of the ice shelf (2017), and the end of our observation



period (2023), when the longer term impact of the unpinning on the distribution of basal channels can be assessed. The transect in Figure 4a is along flow (shown between X and X' in Figure 4b) and intersects areas where the pinning point was observed

to be grounded in both 2011 and 2017. The transects in Figure 4c is across flow (shown between Y and Y' in Figure 4b) and intersects where the pinning point was until 2011. The third transect shown between Z and Z' in Figure 4d is also across flow and intersects the location where Sentinel-1 observes the ice shelf to be ephemerally grounded in 2017. All 3 of these plots also show the depth of the seabed from BedMachine after merging with data collected in (Dutrieux et al., 2014b) along their respective transects. Grey vertically shaded areas show areas along the transects where the ice was grounded according to

MEaSUREs grounding lines in 2011 and Sentinel-1 in 2017. A dashed line is also plotted in Figure 4c and d. These correspond to the depth of the smoothed ice base in 2011 and 2017, respectively, and represent the ice base if channels did not exist.

Despite the overall ice-shelf thinning during this period (Figure 3), the base depth along the along-flow transect doesn't exhibit a straightforward uniform thinning rate (Figure 4a). Downstream of the 2011 pinning point location (the shaded grey region, approximately 6-8 km along the transect), the ice shelf base is shallowest in 2011 and deepest in 2017. The 2011 depth

represents the apex of the downstream channel discussed earlier (and is also visible in the 2011 base depth shown in Figure 4d). By 2017, this section of the ice-shelf thickens significantly as the thick column of ice is advected downstream, filling in the basal channel. By 2023, however, the entire ice-shelf has thinned, and no part of this transect remains in contact with the seabed.

The two across-flow transects (Figures 4c and d) reveal a more predictable relationship between time and thinning. Overall,

the ice shelf base becomes progressively shallower over the observational period. This is with the exception of the basal channel visible in the 2011 ice shelf base in Figure 4d. The contrast between this channel in the centre of the transect in 2011 and the grounding of the ice in 2017 illustrates how the thick column of ice was advected down from the initial pinning point location and changed the geometry of the ice shelf.

Figures 4c and d also demonstrate how basal channels contribute to the persistence of the pinning point and the intermittent

regrounding of the subsequent thick ice column. In both Figures a hypothetical smoothed version of the ice shelf base would not have grounded, instead they would have left a 30-meter deep cavity between the ice shelf base and the seabed. This suggests that without a pronounced channel and keel geometry on the ice shelf base, the pinning point might not have been sustained for as long, and the thick ice may not have periodically re-grounded.

In Figure 4, discrepancies exist between the grounded area predicted by Sentinel-1 DROT methods and the locations inferred

by comparing CS2 ice base data with the modified BedMachine seabed depth. Several factors contribute to this difference. The CS2 DEMs — and consequently the ice draft — are calculated by averaging data points over a full year, which may smooth some of the finer details. Channelised structures are also under-represented on the ice surface due to bridging stresses (Wearing et al., 2021; Drews, 2015) or kilometre scale gradients in ice density (mainly firn air content) (Dutrieux et al., 2013), so our derived ice shelf base depth may not capture the full channel geometry. Despite these limitations, our observations align

reasonably well with those from Sentinel-1.





**Figure 5.** The main variables needed to calculate basal melt rates in the black rectangle in fig 2a. (a) Thickness, (b) Lagrangian change in thickness, (c) basal melt and (d) ice flux divergence. All variables are from 2015.

## 3.2 Channelised Melt

We next use these data to assess the spatial distribution of ice shelf melting at a channelised scale. To do this, we start by focusing on the area shown by the black rectangle in Figure 2a. Figure 5 shows the thickness, Lagrangian thickness change and melt centered on July 2015 and ice flux divergence in 2015. The final three variables, those that are rates, are shown as ice



loss, where a positive ice loss represents thinning and a negative ice loss represents thickening of the ice shelf. At this location in 2015, the magnitude of ice loss caused by melt is highly dependent on the distance from the grounding line, whereas the ice loss from ice flux divergence doesn't appear to depend on the distance from the grounding line. The magnitude of ice loss contribution from basal melt is also significantly greater than the contribution from ice flux divergence ($> 100$ m/yr compared with $\sim 40$ m/yr, respectively).

In the area shown, basal channels are clearly present within the thickness map (Figure 5a). However, it is harder to visually separate the channelised pattern within the thickness change variables (Figures 5b-d). To investigate the channelised pattern within these variables we will now focus on high-pass-filtered anomalies of these fields, which have been calculated by subtracting low-pass-filtered fields (where each grid point is replaced by a 2-D Gaussian average over a radius of 7 km), similarly to Dutrieux et al. (2013). Figure 6 shows the same variables as in Figure 5 but as high-pass anomalies. A positive ice loss anomaly means more thinning and a negative ice loss anomaly shows anomalous thickening.

There are two main channels within this area that can be seen by the negative (blue) thickness anomaly in Figure 6a. They both extend the entire length of the area with little across-flow deviation. The channel on the southern side of this area, indicated by the pink arrow, will be referred to as Channel 1 and the channel on the northern side, indicated by the cyan arrow, will be referred to as Channel 2 from now on. Both Channel 1 and 2 are approximately 2.5 km in width but Channel 1 has an amplitude of 160 m, whereas Channel 2 has an amplitude of only 80 m. Channelised melting can be detected in both of these channels near the grounding line and particularly on the channel's south western flanks, where anomalies are $< -30$ m/yr. The channelised melt pattern becomes less prominent downstream. There is also a group of smaller channels on the southern side of this area which are more variable in distribution. Our method doesn't detect a channelised melt signal within these smaller channels.

Figure 6d displays the ice flux divergence anomaly overlaid with velocity anomaly vectors, illustrating the influence of basal channels on both ice flux divergence and velocity. Ice flow converges into basal channels and diverges away from basal keels. This pattern can be seen right down to the smaller channels on the western side of the area, where channels are just 1 km wide with amplitudes of less than 50 m. This channel-scale distribution of secondary ice flow is consistent with a viscous response to ice thickness gradients (Wearing et al., 2021).

### 3.3 Channelised Melt Evolution

Although previous studies have used DEMs with a higher spatial resolution to derive melt rates (Shean et al., 2019; Zinck et al., 2023), these are less frequently acquired. The strength of CS2 data resides in its long term stability and the frequency of its sampling. With an annual moving window averaging technique, we can observe how channelised features evolve at sub-annual time-scales. Figure 7 shows the elevation (b,f,j) and melt (d,h,l) anomalies along different sections of the ice shelf and how they change in time. As in Figure 6, small-scale high-pass filtered anomalies are taken with respect to a Gaussian 2D filter with a radius of 7 km. The Gaussian filter and therefore the high-pass anomalies are re-calculated at every time step. The first column of Figure 7 shows the time averaged elevation (yellow in a,c,e,g,i,k) and melt (pink in c,g,k) anomalies across the different sections.





**Figure 6.** High-pass filtered anomaly of the main variables needed to calculate basal melt rates in the black rectangle in Figure 2a. (a) Thickness anomaly, (b) Lagrangian change in thickness anomaly, (c) basal melt anomaly and (d) ice flux divergence and velocity anomalies. The black lines on all plots are the zero thickness anomaly contour lines. All variables are from 2015. The pink and cyan arrows indicate Channels 1 and 2, respectively, as discussed in the text.

Figures 7b and d show the elevation and melt anomaly, respectively, across a transect near the grounding line (solid white transect between A and A' on Figure 1). This is in a Eulerian framework, so the transect location is static in time. The location





of this transect is between the main grounding line and the location of the pinning point in 2011. Therefore geometric changes of the ice shelf here have not been directly impacted by the change in pinning point geometry during this period.

Channels 1 and 2 remain stable throughout the entire period, retaining consistent geometries (Figure 7b). Both channels are associated with positive melt anomalies along their apex (minima in surface elevation anomaly correspond to maxima in melt anomaly, Figures 7c and d). The time averaged melt anomalies within Channel 1 and 2 are $\sim$ 30 m/yr and 20 m/yr and the two

highest melt regions correspond to the two lowest elevation regions (Figure 7c).

Figure 7f and h present the elevation and melt anomalies, respectively, along a transect further downstream on the ice shelf (solid white line between B and B' in Figure 1). Similar to Figures 7b and d, this transect is fixed in time. This transect lies about 10 km downstream of the 2011 pinning point location and therefore the elevation and melt rates across it are directly influenced by the changes in pinning point geometry. At the start of the observation period, the channelised geometry is

complex and evolves rapidly. However, by 2015 a persistent channel begins forming in the south of the transect, corresponding to the upstream location of Channel 1. This stabilised geometry likely reflects the impact of the pinning point ungrounding, creating a seamless connection between the main grounding line and this location.

Within the channel, a sustained negative melt anomaly emerges after 2016. Despite the channel's consistent geometry only occurring from 2015, the time-averaged melt anomaly in this area remains strongly negative, and the surface elevation and

melt anomaly are now in phase (Figure 7h). This contrasts with the transect closer to the grounding line (Figures 7c and d), highlighting a regime shift between the near-grounding zone, where Channel 1 is carved, and a few kilometers downstream, where it is eroded. This transition aligns with a theoretical understanding where plume dynamics are more influential near the grounding line (see above), but the depth-dependent freezing point and vertical ocean temperature distribution become more relevant downstream (Dutrieux et al., 2013; Gladish et al., 2012).

We also note that the Coriolis offset in the melt anomaly present near the grounding line is no longer present downstream (Figures 7d and h). This could be a result of the thinner nature of the channels downstream (Millgate et al., 2013) but is more likely due to the aforementioned change in melt regime. Downstream, the keels are melting more than the channels, and therefore melt is not driven by Coriolis-influenced plume dynamics (Gladish et al., 2012).

Figures 7j and 7l offer a Lagrangian perspective, tracking the evolution of the A-A' transect from Figures 7b and d as it moves

downstream on the ice shelf through to 2021. This method allows us to follow the transformation of a single cross-section of ice over time. By late 2020, the transect has reached the dashed white line between points a and a' in Figure 1.

Initially spanning 2 km in width, Channel 1 persists throughout the observational period but undergoes geometric changes. Smaller channels, approximately 1–1.5 km wide, branch from the central, narrow feature. While the primary orientations of Channels 1 and 2 remain relatively stable, these smaller branches appear to veer westward. This branching could result from

preferential melting, carving out new channels from the keels of the parent channels (Gladish et al., 2012). It may stem from the weakening of the overall plume-driving buoyancy force beneath the ice shelf caused by the flatter background ice base slope downstream — perhaps similar to the way a river delta spreads out on flatter terrain. Finally, the balance determining channel scales, an interaction between ocean and ice dynamics, could be influenced by the thinner ice towards the calving front and reduced viscous ice response to melting via secondary flows (Wearing et al., 2021).

**Figure 7.** Elevation and melt anomaly evolution for different sections on the ice shelf. (b) and (d) are the elevation and melt anomalies between A and A' in Figure 1. (f) and (h) are the elevation and melt anomalies between B and B' in Figure 1. (j) and (l) are the elevation and melt anomalies originating from A and A' in Figure 1 in a Lagrangian framework. The transect has been advected forward in time until it is between a and a' at the end of 2020. (b), (d) and (f) are overlaid with the zero contour line from (a), (c) and (e), respectively, in black. The left column of panels shows the time averaged value of elevation and melt from the adjacent panel. The pink and cyan arrows indicate Channels 1 and 2, respectively, as discussed in the text.



Regardless of the mechanism driving this branching, it leads to a near doubling of the channel wavenumber (the number of channels per unit distance) as the transect advects downstream. Gladish et al. (2012) and Millgate et al. (2013) demonstrated that an increase in the number of channels (i.e., wavenumber) near the grounding line is associated with a reduction in overall melt rates. Despite the aforementioned presence of other factors increasing melt at the grounding line, it is interesting to note the correspondence between melt decrease and increased number of channels as the transect progresses down the ice shelf.

Temporal variability in this Lagrangian framework appears noisy and is largely influenced by the spatio-temporal fluctuations in the velocity divergence field. We are currently limited to using annual velocity fields to estimate divergence. This and potentially other methodological limitations make assessing channelised melt evolution in this framework challenging. However, by comparing the channelised melt pattern in Figures 7c and 7g, it is clear that Channel 1 undergoes a noteable transition. Its apex shifts from being associated with a positive melt anomaly (indicating channel calving) to a negative melt anomaly (where the keels melt faster). Despite this regime change, the time-averaged melt anomaly within the channel remains negative (Figure 7k).

    Outside Channel 1, the overall pattern of wide channels splitting into smaller branches persists. However, the melt signal exhibits noisy, discontinuous steps at yearly boundaries due to changes in the divergence field used in the melt calculations. While smaller discontinuations are visible in the melt product of the two Eulerian transects, the combination of changing space and time in the Lagrangian framework emphasises the impact of these discrete steps in the divergence field. This highlights the sensitivity of melt evolution assessments to the choice of velocity divergence field.

## 3.4    Coriolis Favoured Melting

Several studies have hypothesised that melting in channels is enhanced on the Coriolis-favoured side, particularly near the grounding line (Gladish et al., 2012; Millgate et al., 2013; Dutrieux et al., 2013; Gourmelen et al., 2017; Marsh et al., 2016; Cheng et al., 2024). To further investigate the asymmetric distribution of melting across a channel (Figure 7c) and how this distribution varies as a function of distance from the grounding line, we now consider Channel 1 in more detail. This channel has been chosen because it is persistent in time, by the end of the observing period it extends to the calving front (Figure 7f) and we also derive the largest melt signal within it. Therefore, we have the best chance of observing melt variations across it within our data.

To perform this analysis we need to compare observations on the north eastern and south western flanks of the channel at different distances from the grounding line. For each distance from the grounding line, an across-channel transect is drawn and the elevation and melt observations along that transect are used. To divide the melt observations into two flanks, we must first define the boundaries for these areas. The two flanks are defined as the area of ice base between the east and west keels and the channel crest, where these three points are located by extracting the turning points of the elevation anomaly across the transect (where $\frac{dh}{dx} = 0$ and $\frac{d^2h}{dx^2} \neq 0$). The location of the channel crest is then defined as the minimum turning point and the location of the two keels are the maximum turning points on either side. The location of the flank boundaries are recalculated for each month of observations to account for the changing geometry of the channel. Once the areas of the channel flanks along the transect have been defined, all melt observations on either flank are extracted. Once this method has been applied to all months



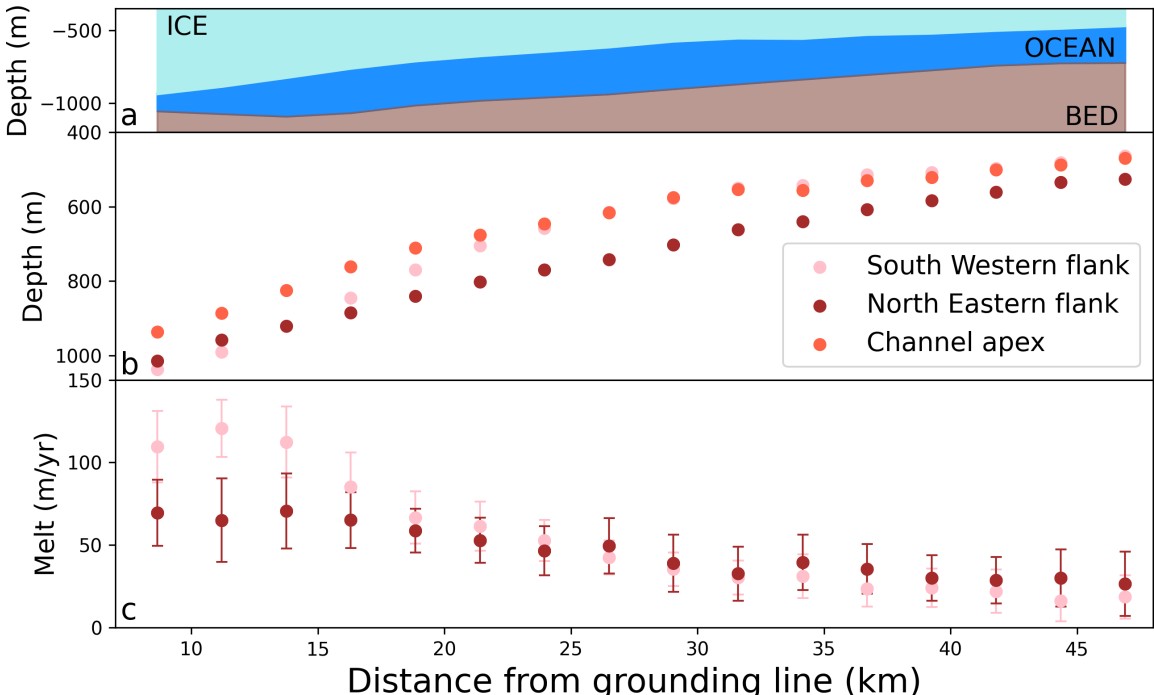

**Figure 8.** Comparison of variables between the east and west flank of Channel 1 as a function of distance from the 2011 grounding line. (a) shows the depth of the channel crest and the depth of the seabed from BedMachine merged with Dutrieux et al. (2014b) observations, (b) shows the average depth of the channel crest and the east and west keel, and (c) shows the average melting on the east and west flank of the channel.

in our observing period, the data is averaged in time. This method is then repeated over the entire set of transects, located at

different distances from the grounding line.

To classify the geometry across the transect as a channel, certain criteria are required from the observations. The channel must be at least 1.5 km in width (3 grid points on either flank for a symmetric channel). If the algorithm fails to define one of the flanks, or if one flank is over 1 km in width longer than the other, then all observations in that month along that transect are discarded. This has been done for two reasons: first, to remove observations where the algorithm has failed to correctly

identify a channel, and second, to ensure there isn't a significant discrepancy in the number of observations between channel flanks. These criteria have been implemented to ensure the algorithm only selects data where the channel is big enough and symmetric enough to compare the melt rates on either flank of the channel. Closest to the grounding line there are over 800 melt observations on either flank of the channel but further downstream there are between 200 and 400. This is because at the start of the time period the channel does not extend to the end of the ice shelf, and the algorithm correctly detects no channel.





Figure 8a shows the main geometry of the ice shelf cavity beneath the channel. The ice shelf is plotted in light blue and the base depth is the channel crest depth, the seabed is brown and its depth is taken from the BedMachine merged with Dutrieux et al. (2014b) observations, and the ocean cavity between them is plotted in dark blue. All subplots here are shown as a function of the distance from the grounding line. In Figure 8b we show the average depth of the channel crest (same as 8a) and those of its east and west keels that have been extracted from the method described above. These are the three locations that act as the

boundaries for each of the channel flanks. The depths of these 3 locations are calculated for every month of observations and the mean of those observations has been shown here. Figure 8c shows the mean melt on the north-eastern and south-western flanks of the channel. As with the depths, melt observations are extracted for every month of observations and the mean of those observations has been plotted. The standard deviation of the two groups is also shown.

Near the grounding line, the depths of the two channel keels are very similar (Figure 8b). However, moving downstream,

the south western flank thins more rapidly than the north eastern flank. By 20 km from the grounding line, the south western flank has almost disappeared, and the channel evolves into a more cliff-like structure. Figure 8c shows the average melt rates on either flank of the channel as a function of distance from the grounding line. Within the first 15 km from the grounding line (approximately 25% of the ice shelf), melt rates on the south western flank are over 50% higher (40–50 m/yr) than those on the north eastern flank. This asymmetry is likely driven by the Coriolis force acting on the ocean plume within the channel,

which enhances erosion of the south western flank. Similar structural evolution of channels has been observed in model results (Cheng et al., 2024). While previous studies have noted this Coriolis-driven phenomenon (Marsh et al., 2016; Dutrieux et al., 2013; Gourmelen et al., 2017) and it has been modelled extensively (Gladish et al., 2012; Millgate et al., 2013; Cheng et al., 2024), we believe this is the first clear observational quantification of the effect.

## 3.5    Channel Formation

To predict how channel geometry — and consequently channelised melting — might evolve in the future, it is essential to understand how channels form. While various mechanisms of channel formation across Antarctica have been observed and modelled (Le Brocq et al., 2013; Gourmelen et al., 2017; Marsh et al., 2016; Alley et al., 2016; Sergienko, 2013; Gladish et al., 2012; Millgate et al., 2013), the formation process on PIG remains uncertain. The persistence of Channel 1 throughout the CryoSat-2 observational period (Figure 7b) provide new insights. This observation suggest a constant formation mechanism

that is closely linked to the grounding zone.

Figure 9 shows the seabed depth (brown) upstream of the grounding line, as measured by airborne radar during the British Antarctic Survey campaign in 2004/05 (Vaughan et al., 2012). Starting in 2011, we advect the location of this flight line downstream with an annual time step. The surface elevation and depth of the ice shelf base as inferred from floatation are then plotted along the advected flight lines. We have only included transects where we are confident the ice at Channel 1 is in

hydrostatic equilibrium (greater than 2 ice shelf thicknesses away from the grounding line).

In the centre of the grounded flight line, there is a $\sim 200$m high feature on the seabed surrounded by smaller ($\sim 50$m high) oscillations. We note that a major part of the ice base evolution is connected to divergence from the shear margins and the associated thinning there, creating a convex glacier tongue with a thicker central line near the calving front. The channelised



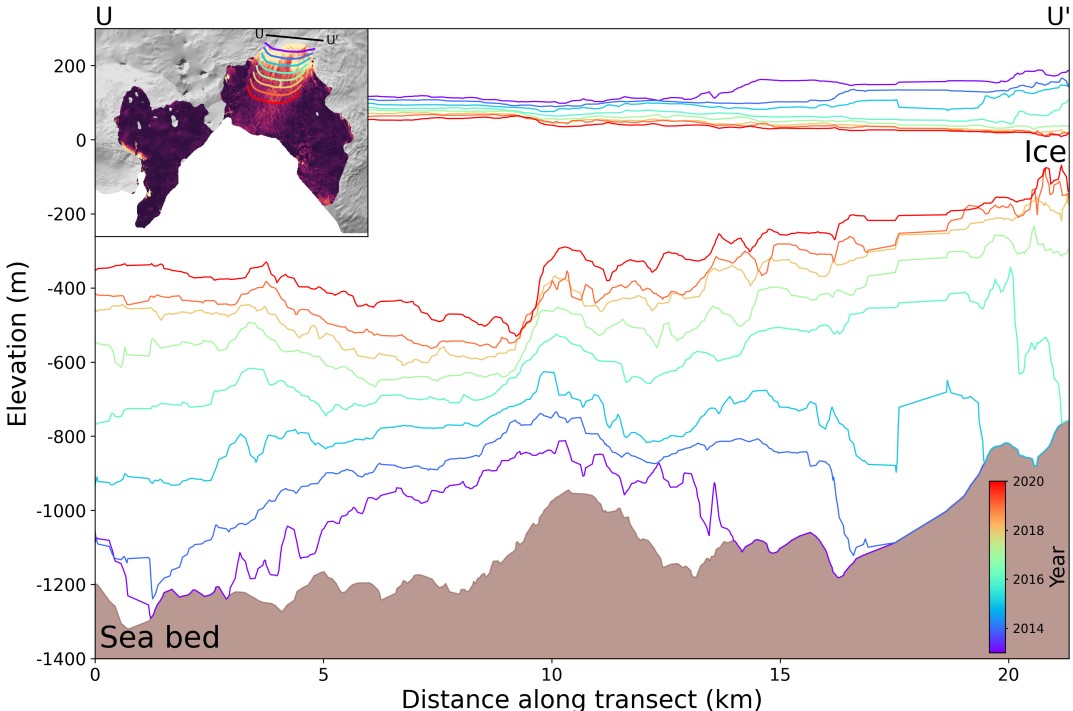

**Figure 9.** The brown area is the ice bed upstream of the grounding line along the black transect on the inset map, as measured from airborne radar (Vaughan et al., 2012). Using ITS_LIVE annual velocity data, the location of this flight line has been advected downstream, with the annual location of these advected transects shown on the inset map. The surface elevation and ice base depth derived from CS2 along each of these transects has then been plotted.

geometry overlays and co-evolves with these large-scale changes. Channel 1 in the centre of the ice shelf appears to inherit
its shape from the large central feature in the seabed. The channel, initially impacted by the presence of the pinning point downstream, begins symmetric (as a reflection of the relatively symmetric bed feature) and gradually evolves to a more cliff-like topography in 2020 as a result of Coriolis-favoured melting, as discussed above (section 3.4, Figure 8). These results suggest that channel geometry is in part a function of upstream seabed geometry on PIG and explains the persistent nature of Channel 1's geometry near the grounding line.

**4   Discussion**

Various approaches have been employed to calculate ice shelf melting using satellite observations (Dutrieux et al., 2013; Gourmelen et al., 2017; Zinck et al., 2023; Shean et al., 2019). In this study, we present the first time series of basal melt rates at sub-kilometre spatial resolution, revealing the time-dependant nature of channelised geometries. On PIG, channelised geometries exhibit variability on annual timescales (Figure 3), and both ice velocity and velocity divergence vary on spatial





length scales consistent with channel structures (Figure 6). Relying on a time-averaged dataset risks aliasing these dynamic features, potentially overlooking spatial variations in melt that contribute to the ice shelf integrated melt rate. Although this impact is especially pronounced on PIG, where abundant basal channels and rapid flow complicate the dynamics, we expect similar considerations to be relevant across other ice shelves. Further investigation is needed to understand the implications of these methodological choices and to determine how they vary among different ice shelves. Ultimately, the method we propose may help refine historical mass loss estimates through better estimating the spatio-temporal variations of melting on sub-kilometre scales.

Furthermore, this methodological sensitivity may affect comparisons between satellite-derived melt rates and in situ observations. Precise consideration of the spatial context of in situ measurements relative to ice shelf geometry is essential. If satellite-derived melt rates alias spatio-temporal features, a meaningful comparison with in-situ data becomes challenging.

The method we have used to estimate basal melt rates at a given time stamp (Section 2) uses elevation data spanning 3 years. All steps of the method use flow-line advection to time-centre the data and account for spatio-temporal changes. This approach addresses errors induced by feature advection through each observing window, but at the cost of introducing temporal smoothing. This creates an effective along-flow smoothing kernel of 3 years (12 km if the ice shelf is flowing at 4 km/yr) but introduces no smoothing across-flow. The method therefore permits shorter wavelengths in the across-flow direction than in the along-flow direction. Despite the disregard of the directional dependence of elevation change, we argue this decision is critical to obtain channelised signals in our dataset.

Our findings lend strong support to theories positing that channelised melt is concentrated on the Coriolis-favoured flank, where rising buoyant plumes are deflected by Earth's rotation. Within 15 km of the grounding zone, we observe that ocean-driven melting intensifies along the south western flank, with melt rates approximately 50% higher than those on the north eastern flank. As a result of this asymmetry, we see the near-disappearance of the south western flank of the channel and the steepening and deepening of the north eastern flank. Despite this, the channel apex does not migrate away from the flow lines (Figure 7j) when temporal variability of ice velocity is taken into account, contrary to suggestions from observation (Alley et al., 2024) but aligned with previous model predictions (Gladish et al., 2012).

In warm cavities such as that under PIG ice shelf, theory shows that we would expect more melting in deeper channels as more vigorous melt plumes are associated with increased friction velocity (stronger currents) and warmer near-ice temperatures (Cheng et al., 2024). It has also been shown that the Coriolis force has a larger impact on distributing melting within wider channels (Gladish et al., 2012; Millgate et al., 2013). In ocean models, once the width of a channel falls below 2 Rossby radii, the Coriolis force plays a smaller role in the momentum balance and the circulation moves away from geostrophy, tending towards geostrophic overturning (Millgate et al., 2013). Our observations, though limited mainly to 2 channels, show some agreement with these theories: the deeper Channel 1 consistently shows more melt, and the wider Channel 1 has a persistent westward leaning melt distribution, with its western keel displaying positive melt anomalies and it's eastern keel displaying negative melt anomalies (Figure 7).

Despite the agreement of our observations with theories of narrow channels being less impacted by Coriolis-favoured melting (Millgate et al., 2013), Cheng et al. (2024) showed that if one assumes a symmetric and unchanging channel geometry across



the whole length of the ice shelf, Coriolis-favoured melting persists along the whole length of the ice shelf due to the ice-ocean boundary current increasing in velocity. These theories, combined with observations, suggest the distribution of melting is determined by a complex interplay between plume dynamics, channel and ice shelf wide geometry-driven ocean circulation, and the presence of other forced ocean currents (e.g. tides). Further research is needed to better understand how this asymmetry varies spatially and temporally across different channels and ice shelves.

Previous studies have shown that the largest errors in deriving melt rates from remote sensing data occur along channel walls, where bridging stresses across ice-thickness gradients cause the ice shelf to deviate from hydrostatic equilibrium (Drews, 2015; Wearing et al., 2021). Bridging stresses mean that the channel is actually deeper than inferred from surface topography and it is expected that melt is underestimated at the channel apex as a result (Wearing et al., 2021). While such errors may affect the quantitative melt rates derived, we do not expect them to influence our qualitative conclusions, for example regarding

Coriolis-favoured melting, as the errors should at least initially be symmetric across the channel.

The overall contribution of basal channels to ice-shelf stability remains a subject of debate (Alley et al., 2022). Our findings confirm a link between the existence of channels and the ability of their keels to create ephemeral pinning points. By focussing melt within channels, deeper keels form than if the ice melted uniformly. This increases the likelihood of grounding. By enhancing the persistence of pinning points, basal channels may have indirectly delayed the loss of buttressing capacity on

PIG. This observation aligns with previous research that highlights the role of basal keels in the intermittent regrounding of pinning points (Joughin et al., 2016). Alternatively, transient pinning could act as a focal point for localised ice strain, potentially leading to crevassing and rifting in areas with strong thinning gradients (Arndt et al., 2018). While our results do not provide a definitive answer on this point, nor do they quantify whether pinning points form or persist due to the advection of basal channels around Antarctica, they highlight the complex role of channel-pinning point interactions in ice-shelf dynamics.

The formation mechanism of basal channels varies significantly across Antarctic ice shelves, with key implications for ice-shelf stability and melt patterns. Notably, the primary channel on PIG retains its form from upstream of the grounding line, where bed elevation variations directly imprint onto the ice shelf base. The coupling between the bed and the ice shelf base determines the spatial distribution of basal melting, concentrating the highest melt rates within the Channel 1. These findings suggest that channel distribution is closely linked to the underlying bed geometry, and therefore, changes in grounding line

position would likely shift the distribution of basal channels and basal melting. This highlights the dynamic nature of basal channels as both products and regulator of ice shelf geometry and stability.

## 5 Conclusions

This study analyses the spatial and temporal changes in channelised ice shelf melting on Pine Island Glacier between 2011 and 2023 using CryoSat-2 surface elevation data. We have demonstrated that it is possible to use these data to derive monthly

DEMs and melt-maps at a 250 m spatial resolution over Antarctic ice shelves. Furthermore, we emphasise the importance of incorporating accurate velocity divergence field when calculating ice shelf basal melt rates on channel scales from satellite data. This precision is crucial for capturing small-scale spatial changes, as high spatial frequency melt and thickness gradients signif-



icantly impact small scale ice velocity divergence anomalies. These methodological conclusions have important implications for future satellite observations of ice shelf basal melt, potentially enhancing the accuracy of historical ice loss estimates.

Using the data presented here, we have shown the evolution of basal channels on PIG, emphasising their interactions with pinning point dynamics and their role in modulating basal melt rates. The main channel on PIG inherits its shape upstream of the grounding line from an ice bed hill. Once floating, ocean melt is concentrated on its Coriolis-favoured flank. Within 15 km of the grounding line, melting on the Coriolis-favoured (south western) flank of a channel can be over 50% greater than on the north eastern flank. This channel, and its corresponding keel, contribute to the formation of the central pinning point on PIG.

This channel geometry facilitates ephemeral re-grounding as it moves downstream, potentially influencing ice-shelf stability.

*Code availability.*

The code used in the melt calculation can be found here: https://github.com/katiel1108/PIG_melt.

*Data availability.*

We are currently working with the Polar Data Centre to create a DOI for the DEMs and melt maps created and used in this 470 study.



**Appendix A**

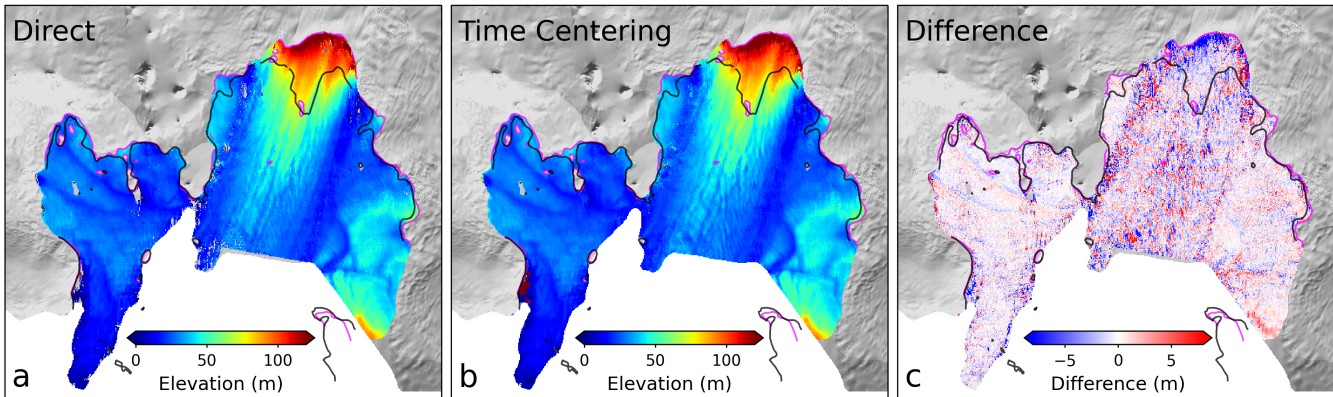

**Figure A1.** PIG DEMs in 2015 generated by (a) direct binning and (b) time-centering. (c) is the difference between (a) and (b).

*Author contributions.*

KL, PD and PH planned the research; KL conducted the data analysis, except the Sentinal-1 DROT data which were processed by BJW. KL wrote the manuscript. All authors provided insights on data interpretation and reviewed and edited the
manuscript.

*Competing interests.*

We declare that the authors have no competing interests.

*Acknowledgements.* This work was led by Katie Lowery at the British Antarctic Survey (BAS) and the School of Earth and Environment at the University of Leeds. Katie Lowery was supported by the Natural Environment Research Council (NERC) Satellite Data in Environmental
Science (SENSE) Centre for Doctoral Training (grant no. NE/T00939X/1). This research was supported by OCEAN ICE, which is co-funded by the European Union, Horizon Europe Funding Programme for research and innovation under grant agreement Nr. 101060452 and by UK Research and Innovation. OCEAN ICE Contribution number XX. The authors gratefully acknowledge the European Space Agency (ESA) and the European Commission for the acquisition and availability of Copernicus Sentinel-1 data. Benjamin J. Wallis was supported by the Panorama Natural Environment Research Council (NERC) Doctoral Training Partnership (DTP), under grant NE/S007458/1, and the UK
EO Climate Information Service, under grant NE/X019071/1.



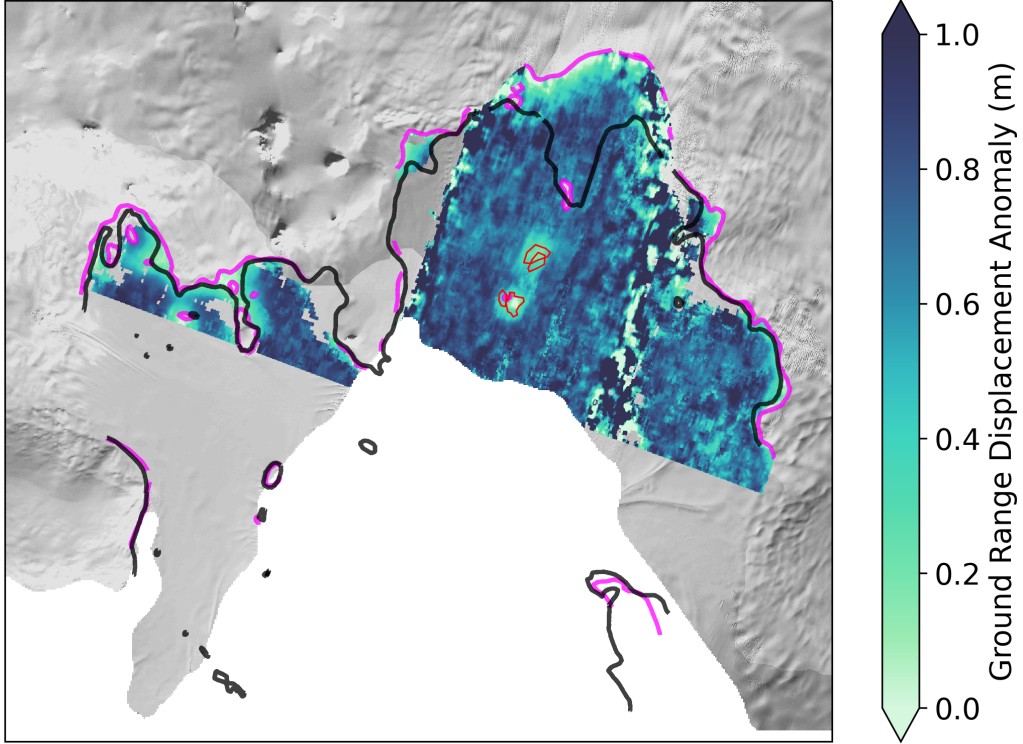

**Figure A2.** An example of the ground range vertical displacement anomaly as calculated by applying the DROT method to Sentinel-1 data. The red contour lines show the manually delineated areas of grounded ice. The data shown is calculated by comparing acquisitions from 2017/01/27 and 2017/02/02 with respective tide heights of 0.65 and 0.11 m.

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
