# Peer review of "Spatio-temporal melt and basal channel evolution on Pine Island Glacier ice shelf from CryoSat-2."

_EGUsphere, 2025_

## Author Comment (AC1)

Comments from the reviewer are given in black.

Author responses are in red, and **proposed changes or amendments to the manuscript are given in bold red.**

**RC2 – 'Comment on egusphere-2025-267', Viet Helm**

General comments

In this study a new approach to calculate ice shelf basal melt rates on monthly resolution for Pine Island Glacier is presented. The study is a case study to demonstrate the new approach which uses CryoSat-2 swath data to derive monthly melt maps with a 250 m spatial resolution. The results show, that near the grounding line basal melting is 50% stronger on the western flank of a channel that on the eastern flank. In addition, findings suggest that the channelized geometries on PIG are triggered upstream and that the channel geometry facilitates ephemeral re-grounding as it moves downstream, potentially influencing ice-shelf stability. In general, the paper is well written, clearly structured and the presented idea to derive melt rates is novel and worth to be published in TC. Equations used are correct and figures are of good quality while mostly supporting the analysis.

We would like to thank the reviewer for their detailed comments on the manuscript and the expertise they have brought to the review. Following their comments, the manuscript has been greatly improved.

While reading I think that the results part is very voluminous and already includes sections which should be more presented in the discussion section (e.g. L202 – L225). Please check carefully.

Thanks to the reviewer for highlighting this – **we will distil the results section and move any points to the discussion where appropriate.**

Shean (2019) used high resolution Tandem-X DEMs to derive melt rates of PIG. They found a higher melting associated with basal channels and deep keels near the grounding line and relatively shallow keels over the outer shelf and do not discuss a more pronounced western flank melting. It would be nice and important to see if you can confirm and discuss the findings of Shean in more detail and also if this Coriolis dependent melting as it was not discussed in Shean (2019) might be a result your processing methodology.

Thanks for highlighting the need for more discussion on this point. We are only able to deduce the Coriolis-favoured melt pattern because of the large number of observations within our dataset. Compounding and averaging these observations allows us to detect such a signal over a relatively small spatial scale within relatively noisy data. Shean (2019) didn't comment on Coriolis-dependent melting. Their dataset only contained a single melt map, which spanned 2008-2015 and was calculated by averaging all melt rates that intersect a given pixel in the time period. It is therefore likely that either they didn't have enough data to detect this Coriolis signal, or this signal was aliased by the composite method used. Nevertheless, it is interesting that this signal was not reported within their dataset. **We will add a discussion point on this to the revised manuscript.**

As a general remark. How much is the velocity changing throughout the observed time period? Can you please make a figure in the supplements showing the difference of the x and y

components of each single velocity field to an averaged velocity. To my opinion it would be much better to use an averaged field in the whole processing as long as the velocity is more or less constant throughout the time period. This would substantially minimize errors in the Lagrangian shift, which is based on pixel to pixel shifts and therefore very sensitive to noise of the velocity components on the pixel scale. I think this is important, especially as you try to analyze small scale melt differences within a channel.

In the creation of this manuscript, we considered the methodology used in great detail. Figure R1 shows the difference in each annual velocity field from the time averaged field. The velocity has increased by ~1000m/yr during this time, and the flow diverted westward after 2018. Due to these changes, we don't think an averaged velocity field would be appropriate here. **We will add further discussion on this in the manuscript and include a supplementary figure highlighting how the velocity of PIG has changed during this period.**

[Figure]

***Figure R1**: Velocity change with respect to the time-averaged field during the observed period.*

Suggestion for all overview figures: I think the western part of the ice shelf doesn't need to be included as it is never discussed in the paper. Please zoom in to the main fast flowing part which you are focusing on (Fig 1, Fig2, Fig3, Fig4b, inlet Fig9, Fig A1, A2)

Thanks for your comments on the figure layout. While the western part of the ice shelf isn't directly discussed in the manuscript, the area has been highly dynamic during this period and significant changes have been observed. We therefore argue that keeping this in the figures is both useful for orienting the reader and of interest to some readers.

In addition, the choice of the selected along and across flow segments which are varying in the paper make it difficult for the reader to follow. Can you please include in your overview Fig1 also the cross section X-X', Y-Y', Z-Z' and the box S-S' and U-U'

This was something we tussled with in the creation of the manuscript. We acknowledge that the number of different sections and transects is a little involved and requires the reader to take

special care. However, when we drafted Figure 1 with all sections overlaid (Figure R2), we think it looks messy and makes it harder for the reader to decipher.

[Figure]

**Figure R2:** *Example of Figure 1 with all transects marked on it.*

Z-Z' seems to be very similar to B-B' – why not using the same.

**We will make these transects the same.**

Can you please slightly enlarge S-S' that it includes Y-Y', Z-Z' and B-B' and please also mark in Figure 5 and 6 the position of those cross sections. This would help to understand better otherwise it's a bit confusing.

Thanks for highlighting this. **We will enlarge the area of S-S' in figures 5 and 6 and mark on the transects.**

Please don't use rainbow color scale for elevation and thickness.

**We will change all rainbow colour maps to something more appropriate.**

here you claim that the new method can be applied on every ice shelf or terrain. I think this is not correct. You still need sufficient data coverage of the swath data to form this DEMs based on 1 year of acquisition data. And this is not the case for most of the ice shelves. For PIG or Dotson it seems to work.

Thanks for highlighting this. **We will rephrase this statement and add the caveat that this is subject to data coverage.** However, others have shown similar methods can work on other ice

shelves: Getz (Wei et al., 2020) Crosson (Lilien et al., 2019), Thwaites (Gourmelen et al., 2025), Totten (Gwyther et al., 2023), Larsen B (Surawy-Stepney et al., 2023).

Usually from theory quasi nadir swath processing is affected by phase ambiguities and low coherence over flat terrain, where across track slope is less than half the antenna beamwidth – which usually is the case over ice shelves. As Cryosat2 has a small mis pointing of 0.1° this left and right looking phase returns are not completely canceling, allowing to detect some coherence and therefore to derive elevation estimates across track in some places like PIG. I think it is important to be mentioned.

Thanks for raising this point. The swath processing does rely on the presence of a surface slope to be viable, and for very low slopes, a left-right ambiguity can occur and usually results in reduced signal quality. This has been addressed in the response to reviewers in Gourmelen et al., 2025. While a number of previous studies have shown we can retrieve swath elevation for a significant portion of CryoSat-2 waveforms over ice shelves (Gourmelen et al., 2017; Wuite et al., 2019; Lilien et al., 2019; Goldberg et al., 2019; Wei et al., 2020; Davison et al., 2023; Gwyther et al., 2023; Surawy-Stepney et al.,, 2023), we hypothesize that it has to do with ice shelf heterogeneity in backscattering properties across ice shelves, and to the fact that CryoSat-2 is not pointing perfectly at nadir but is mispointing slightly (Recchia et al., 2017). **We will add a comment on this to the methods section of the manuscript.**

Here you argue that the time centring method is more accurate than just binning and compare both DEMs in A1. How do you know which one is better? Of course you reduce the averaging of across flow features but with the time centering method you also introduce errors which are related to the velocity field and it's derivative, which I assume is very noisy when using yearly velocity fields (see comment above). I would suggest to compare to other high resolution DEMs like the Tandem-X DEM presented in Shean (2019) to evaluate which method is better.

Thanks for raising this point. We have completed a comparison with a Shean (2019) DEM to confirm that the time-centering method is better. Figure R3 shows a histogram of the elevation difference between the Shean (2019) DEM and our time-centered DEM and our direct DEM. The average difference between the time-centred DEM and Shean is 0.67m, whereas the difference between the direct DEM and Shean is 1.37m. **We will add the average difference shown in these plots to Figure A1 in the paper to better illustrate which method is better.**

[Figure]

***Figure R3***: *Difference between our time-centered DEM and direct DEM with Shean (2019) DEM.*

I would also like to see a typical point cloud coverage of one month of swath elevations to get an idea of the general data coverage.

See figure R4 for an example of a month's worth of point cloud coverage.

[Figure]

**Figure R4**: *Example of one month of CryoSat-2 swath coverage from 2018/01.*

As you use the median as a very robust averaging method, I would like to see the standard deviation for each pixel to get an idea of how much the swath elevation point cloud elevations are varying within one pixel.

Figure R5 shows the standard deviation for our 2018/06 DEM.

[Figure]

*Figure R5*: *2017/06 DEM Standard Deviation*.

Do you filter or exclude outliers of the swath data before averaging?

Swath returns with a coherence below 0.8 are masked. **We will include this in the methods section of the revised manuscript.**

Can you please exactly explain where H is coming from? Due you use the mean(h) of both contributing DEMs, or h of the first DEM or a constant h for all time steps?

H is from the first DEM. **We will clarify this in the revised manuscript.**

Is the noise of the monthly melt maps maybe related to the noisy velocity field?

It is possible that the noisy velocity fields are contributing to the noise in the monthly melt maps. It is also likely that some of the melt map noise is coming from noise in the CryoSat-2 DEMs. Despite this, we think that remaining with the slightly noisier velocity fields is more appropriate that using a 'cleaner' time-averaged product that doesn't capture the evolution of the ice shelf during this time (Figure R1).

The averaging of monthly data to reduce noise is a well know procedure. However, I do not understand why you advect the melt maps. To my opinion this is not correct. The overall melt pattern is not moving in a Lagrangian sense with the ice. The melt is triggered by ocean water masses and the ice is moving. This means that the melt pattern can locally change with time due to changes in warm water supply through the ocean but this warm water supply is decoupled from the ice movement. Therefore, an averaging of monthly melt maps should be done without advection. I also think because of this additional advection you change the melt distribution across a channel. And this change is correlated to the across flow component of the velocity field. Therefore, I would like to recommend to redo the analyses based on not

advected averaging and see new figures 6,7,8. Will this change your conclusion of pronounced western flank melting and support the findings in Shean of high keel and channel melting in areas close to the grounding line?

As mentioned, please redo the analysis with not advected averaged melt maps.

Thanks for your suggestions on this. While the point raised by the reviewer is correct, ocean induced melting also depends on the ability of the circulation to bring heat to the ice-ocean interface. In a plume context where vertical entrainment is associated with ocean currents, this typically means that high currents are associated with higher melt. And we understand that the ice geometry impacts the ocean circulation (and currents), both at the large and the channel scale. This means that ice movement (and movement of the associated channel geometry) matters for ocean currents, and therefore matters for the spatial distribution of melt. A temporal averaging that doesn't take this fact into account would in effect smooth any channelised melt anomaly that is not aligned with ice flow (i.e. all transverse channels and even the somewhat wiggly longitudinal ones). **We will add a more detailed justification of this choice to the manuscript.**

Please also give the equation of how you derive ice shelf base showing in Figure 4.

**We will include this equation in the manuscript.**

This is an interesting finding, that you are able to see an un- and regrounding in the CS2 data. Can you please confirm that DROT also see an ungrounding of section Z-Z' in 2023?

Yes, the DROT data also shows an ungrounding in 2023. **We will add a comment on this in the revised manuscript.**

In Fig 4b you only show DROT grounding of 2017. I would suggest to zoom in in Fig 4B to only show the relevant ice shelf section and enlarge labelling in 4B. Maye provide another figure like 4B with DROT grounded areas of 2023 to further support the CS2 data.

Thanks for highlighting this. **We will zoom in on figure 4b to only show the relevant portion of the ice shelf and enlarge the transect labels.** As mentioned above, the DROT method doesn't detect any grounded ice in 2023.

As suggested. Why not use a constant averaged less noisy velocity field to avoid effects of changing and noisy divergence.

Please refer to the response above and Figure R1 regarding the change in velocity over the observational period.

Could you please include in your analysis and Figure 8c the melt rate within the channel apex as well as the melt rates of the western and eastern keel. It would be interesting to see if this Coriolis effect, as proposed, is only dominant in the channel and if in the neighboring keels a different melt rate is observed. Furthermore, this would also show whether Shean's observations of higher melt rates at the Keels can be confirmed.

Thanks for this comment. In response to this, we have calculated the melt rates within the channel and keel on the west and east of the channel i.e. splitting the channel into 4 transverse sections as opposed to the 2 (west/east) in the current manuscript. The results show a gradient

between the 4 sections, with the highest melt near the grounding line remaining on the western keel, followed by the western apex, eastern apex and then eastern keel. **We will change the analysis in the revised manuscript to contain this analysis across 4 sections.**

These results still don't support Shean et al (2019) observations of highest melt rates on the keels. However, Shean et al (2019) defined a keel to be where the ice shelf surface elevation anomaly is greater than 1m. This corresponds to a thickness anomaly of ~10m and therefore a base depth anomaly of ~9m. It is therefore likely that, through this definition, melt rates on the channel flank are being labelled as channel keel. **We will include a discussion detailing this mismatch in the revised manuscript.**

Can you please mark with a grey bar the position and extend of the channel you discuss in L379 ff

**We will mark the extent of the discussed channel on Figure 9.**

---

## Author Comment (AC2)

Comments from the reviewer are given in black.

Author responses are in red, and **proposed changes or amendments to the manuscript are given in bold red.**

**RC1 – 'Comment on egusphere-2025-267'**

General comments

This study uses averaging of Cryosat-2 data to produce an 11-year record of centered-in-time annual DEMs for Pine Island Glacier, and then uses these to investigate the evolution of basal channels, primarily through a Lagrangian change analysis. Overall, I find the methods to be sound and the conclusions to be well-supported. I was particularly interested in the relationship between the pinning point and channel evolution. I think this will be a valuable addition to the literature on DEM methodologies, basal channels, and PIG.

We would like to thank the reviewer for their detailed comments and express our appreciation for the deep knowledge and expertise they brought to the review. As a result of these comments, the manuscript has been greatly improved.

If I've understood correctly, anomalies in elevation and basal melt rate are calculated by using a 7 km Gaussian filter, and then subtracting the results from the unsmoothed values. So, this is a neighborhood operation, and these anomalies are saying whether the ice base and melt rate are higher or lower than nearby points. For melt rates, as is described in lines 239-240, "A positive ice loss anomaly means more thinning and a negative ice loss anomaly shows anomalous thickening." It is *technically* correct to use this language; a negative ice-loss anomaly means thickening in the same way that a slightly weaker northerly wind is a southerly wind anomaly. However, the language is misleading, because based on the map in Figure 5, and from what we know about the PIG system, there should be no freeze-on occurring. So, in fact, there is no actual thickening due to melt (freeze-on). The anomaly values themselves are also not particularly meaningful, since they are comparisons to a fairly arbitrary section of ice-shelf base.

What I think *can* be said based on the anomalies is that, first of all, they are really only relevant if you're interested in the relative evolution of a distinctive basal feature such as a basal channel, and that a negative anomaly within the channel indicates that the amplitude of the channel is decreasing, i.e. the channel is filling in. That's interesting and important, but is currently obscured by the presentation in the manuscript.

Many thanks for the comments made here. Melt anomalies identify melt-induced, channel-scale thinning and thickening with respect to the larger scale. As pointed out by the reviewer, this does not necessarily imply freeze-on, and we are careful not to suggest that in our initial submission. However, we agree that the clarity of the manuscript will be improved by incorporating these points. **In the revised manuscript, we will change the wording in lines 239-240 and elsewhere in the manuscript to clarify what is meant and what can be deduced by the positive and negative thickness anomalies.**

I think this can probably be addressed by being more careful with language within the current structure of the manuscript. However, I also think that a careful look at the results presented will reveal that not all of the discussion is relevant to the conclusions. Ideally, I think the results could be shortened quite a bit after reconsidering the meaning of the results and linking them to the conclusions.

Once we have rephrased the terms above, **we will remove parts of the results and discussion that are no longer relevant.**

The length scale over which ice-flow divergence is calculated should be explicitly stated and justified.

Thanks for highlighting that this is missing from the manuscript. **We will add a description to the manuscript to address this. Ice-flow divergence is calculated over a 480m length scale using a centred gradient method.**

In general, there is little to no treatment of error. While I don't know that a rigorous treatment of error is necessary, some discussion at least seems warranted. For example, calculations are made right up to the 2011 grounding line. It is not established how the grounding line has evolved after 2011 and throughout the record developed here. If it has retreated substantially, then using the 2011 grounding line is conservative and probably okay. If it has retreated slowly or not at all, then the measurements close to the grounding line are likely substantially influenced by not being fully in flotation, and these need to be discussed in the results and/or removed from the analysis.

Thanks for highlighting the need for a discussion around the treatment of error within the manuscript. **We will add a paragraph to the discussion about errors within the study.** Throughout the manuscript, we use a cautious 2011 grounding line for all analysis and datasets. The grounding line has retreated by a few kilometre's during this period. While it remains that data near the grounding line is not fully floating, and therefore the assumptions of the methodology don't hold, all basal channel analysis supporting our conclusions is complete on areas of the ice shelf over 2 ice thicknesses downstream from the 2011 grounding line and therefore in an area of the ice shelf we consider to be fully floating.

I'm also concerned that there appears to be no temporal interpolation of velocity fields during point migration – rather, it sounds like the velocity of a migrated point is pulled directly from the annual ITS_LIVE velocity grid for the year into which the point fits. On an accelerating ice shelf, this can introduce large errors in migration location, which should probably be quantified. If the ice shelf is accelerating fairly steadily, linear interpolation between velocity grids would reasonably solve this issue. If I have misunderstood the method, it would be helpful to clarify the text.

Thanks for this methodological suggestion. The integration of velocity variability is something we have considered in great detail and is an involved part of our methodology. The reviewer has understood correctly that the velocity of a single point is pulled directly from the annual dataset corresponding to the year the point is in and is not interpolated in time using annual mean fields. For all point migration, we recalculate the velocity of the point on a monthly time step to account for changes in flow direction and speed as the point moves down the ice shelf. However, we have not quantified the error that ignoring sub-annual velocity changes induces.

On the creation of the manuscript, we made a conscious decision not to interpolate velocity fields. This decision was pushed by our goal to study/quantify basal channels and how they change in time, and we surmised that the velocity and velocity divergence both spatially vary on channel-sized (sub-kilometre) scales. Often, these changes are induced by the presence of a basal channel. Temporal interpolation between annual velocity estimates would alias channelised velocity patterns. This will be especially prominent in cases where basal channels are not aligned with flow.

Over the course of our observing period, PIG flow has diverted west-ward (Figure R1) whereas the orientation of the basal channels discussed in the manuscript have remained consistent (Figure 7). We therefore think interpolated velocity fields would alias the signal we are trying to detect and would not be appropriate.

[Figure]

***Figure R1:*** *Annual velocity difference with respect to the 2011 velocity field.*

Having said that, your question prompted us to re-calculate melt rates using interpolated velocity fields to evaluate the methodological impact. Since the interpolation that we will use is linear, we expect it will have a smoothing effect on the result. However, this will likely be a more robust method for future implementation. **Unless we find anything unexpected, we will use this method in the revised manuscript.**

"A dashed line is also plotted in Figure 4c and d. These correspond to the depth of the smoothed ice base in 2011 and 2017, respectively, and represent the ice base if channels did not exist." I'm not convinced of this equivalency – it's just a smoothed ice base, not necessarily one devoid of influence from channels.

Thanks for highlighting the ambiguity in our language here. Of course, this approach to creating a smoothed ice base and one void of basal channels is simplistic. The smoothed geometry is what we might expect if melt only resulted from ice-ocean interactions over a smooth ice base interface rather than one where the melt distribution is modulated by channel geometries. Of course, this is a simplistic assumption, as we would expect the channelised geometries to modulate melt efficiently. However, this analysis is merely trying to make the point that channel keels gave the ability to touch the seabed whilst a geometry without keels may not. **We will reword this sentence in the manuscript to clarify what we mean here.**

The use of directions is confusing. The abstract uses east and west, lines 242-243 use north and south, and the description of Coriolis-influence melt uses northeast and southwest. This needs to be standardized. I realize that PIG is not directly N-S aligned, but since we normally think of ice as flowing from south to north in Antarctica, I'd personally stick to east and west.

Thanks for highlighting this inconsistency. **We will change these in the manuscript to consistently be east and west.**

"...and the surface elevation and melt anomaly are now in phase." I'm not sure what the point of this is, physically speaking. It's true that the graphs in Figure 7 go in the same direction at this point. But a negative melt anomaly means less melting, which would lead to relatively thicker ice, while a negative ice-thickness anomaly means thinner ice. So, physically, they are at odds with each other. While this is a mathematically correct statement, it doesn't seem to help much with the physical interpretation

We agree with the reviewer that this statement needs some clarification. Physically, the two lines being in phase means that, on average, there is less melting within the basal channel and more melting of the keels. In a steady state system, this means we would expect the channel to decrease in amplitude. This is interesting because it is the opposite of what is seen across the transect closer to the grounding line. **We will add a description to the manuscript that explains what this means physically and why it is interesting.**

Line 281: "...where Channel 1 is carved, and a few kilometers downstream, where it is eroded." These terms don't make much physical sense. Carving is okay, but I assume the term "eroded" is supposed to mean where the channel is filling in (as the opposite of "carved"). Since erosion typically refers to taking material away, this would be a very confusing way to say it

Thanks for the comment. **We will add change the above statement to say 'where it's amplitude decreases through the erosion of it's keels' to increase the clarity of this explanation.**

Similarly, line 309 "indicating channel calving" – I assume carving was meant? Calving would not make physical sense here.

Thanks for spotting this mistake. **We will change this to carving.**

Figure 7 and the text show both Eulerian measures of change and a Lagrangian measure, but it is unclear what differing conclusions can or should be drawn from these differing analysis styles, and it is not brought into the discussion/conclusions. I do think there are some interesting implications about the persistence of a channel imprinted on an ice parcel vs. the influence from Eulerian features such as plumes and pinning points that are hinted at in the manuscript, but could be stated much more explicitly.

Thanks for highlighting this. The Lagrangian measures allow us to track the evolution of a channelised section of ice throughout the observational period – gaining insights into the development of channels as they are advected downstream, their lateral movements and the associated melt rates. As pointed out, these discussions are somewhat different from those drawn from the Eulerian approaches, which instead focus on static sections of the ice shelf. **We will emphasise these differences in the manuscript.**

I believe these features are discussed in some detail in Bindschadler et al. (2011), https://doi.org/10.3189/002214311797409802.

**We will include this citation in the revised text.**

Figure 8 shows a few spots where the algorithm has marked the western flank as being shallower than the apex. I suggest that the algorithm is not working well in these cases, and perhaps these points should be excluded.

Thanks for highlighting this. **We will remove any points where the algorithm detects the keel to be shallower than the channel.**

Sergienko (2013) is a modeling study that clearly shows deflection of channels from flowlines.

**We will include this citation in our discussion.**

West Antarctica abbreviated as WA: Since this is only used a small number of times in the first paragraph of the intro, it seems better to just write out "West Antarctica"

**We will remove the abbreviation in the revised manuscript.**

"data" should be plural throughout – e.g. line 86: "The 12 months of data that *are* required to *create* a single DEM *are* centred around…" (note also that "created" in this line should be "create"

Thanks for pointing out this grammatical error. **We will change this in the manuscript.**

Line 195: "transects" should be "transect"

**This will be changed in the manuscript.**

Compound adjectives or nouns used as adjectives should technically be hyphenated – e.g. "kilometre-scale gradients" in line 223. Even terms like "sea-level rise" and "ice-shelf stability" should technically be hyphenated, although I realize that this isn't necessarily in style. If two nouns are not used as adjectives, they should not be hyphenated – e.g. remove hyphens from "ice-shelf" in lines 206 and 207

**These will be changed in the manuscript.**

Please replace rainbow color maps, which introduce false perceptions of high gradients, with a visually consistent color scheme

**We will change all rainbow colour maps in the manuscript to more appropriate colour maps.**

In general, it's helpful if figure captions stand mostly on their own, so that a reader glancing through the paper can make sense of what you're doing. To that end, it would be helpful to define acronyms within captions (e.g. DROT in figure 1), and although it's fair to say that basemap and grounding lines are the same as in figure 1, it's a pain to go back through to find figure 1 to figure out when the basemap and grounding lines are from. Perhaps consider a legend within the figure for the grounding lines at least, and/or put the date of the basemap in the captions.

Thanks for pointing this out. **We will change all figure captions so that they stand alone, including defining acronyms, giving the dates of the basemap and adding legends for grounding lines.**

Figure 3: The color bar appears to be very saturated, which limits the amount of information we can get from the figure.

**We will expand the colour bar in the revised manuscript.**

Figure 4 (discussed in text): I wouldn't call those lines light blue, blue, and dark blue – at very least the first line is much more green than blue.

**We will refer to the colours as cyan, blue and dark blue in the revised manuscript.**

Figure 5: Titling this figure "The main variables needed to calculate basal melt rates" is odd when one of those variables is basal melt rates themselves. Although using "ice loss" as the color bar title is technically correct, it's a little confusing when on one color map it refers to basal melt (ice is completely lost and turned into water) and on another it's divergence (ice is lost from the pixel but just moves next-door). Consider "ice-thickness loss" instead, here and in the text.

Thanks for highlighting the need for more clarity here. **We will remove the title and change the colour bar label to 'ice-thickness loss' in the revised manuscript.**

Figure 5 (described in text): "basal channels are clearly present within the thickness map." It would be helpful to mark them.

**We will add the same cyan and magenta arrows to Figure 5 as are in Figure 6.**

Figure 7

First, I'm not sure why the time-averaged elevation anomalies (and please clarify in the caption that these are anomalies) are plotted twice; I think it would either be better to plot them once on the same graph with the melt, or to plot the melt and elevation on separate graphs, but I spent a while trying to figure out why those were different.

Please also move the legend so it does not cover up some of those data.

Consider moving those averages to the right-hand side, since they're calculated from the Hovmöller data (just seems more logical, but this is a minor preference).

The caption statement "(b), (d), and (f) are overlaid with the zero contour line from (a), (c), and (e), respectively" has me confused. The zero contour line should come straight out of the Hovmöller, without need for averaging, so I'm not sure why it's linked to the averages. There's also what appears to be a zero contour line on the rest of the Hovmöllers, which aren't mentioned.

It would be easier to spot your cyan and pink arrows marking the basal channels if they were always on one side – probably on the right, since that's where the channels are consistently seen. It would also be helpful if the same locations were marked on the melt diagrams in d, h, and l.

Thanks for the detail in which the reviewer has considered Figure 7. The time-average elevation anomalies are plotted twice to help guide the reader. Having them plotted alone alongside the elevation anomaly hovmoller first presents what these look like, and then keeping them on the second plots alongside the time average melt anomalies allows the reader to interpret the melt

anomaly with respect to the elevation anomaly (interpret the melt anomalies within a basal channel context).

**We will move the time-averaged plots to the right hand-side.**

We apologise for the mistake in the figure caption regarding contour lines and thank the review for noticing this. The caption should read 'd), (h) and (l) are overlaid with the zero-contour line from (b), (f) and (j), respectively, in black.' and **will be changed in the revised manuscript.**

**We will move the channel markers to the right hand-side of the Eulerian plots but keep them on the left hand-side on the Lagrangian plot.** We have decided to keep the markers on the left for the Lagrangian plot because the channels evolve and split up by the time they reach the right-hand side of this plot and therefore marking a single location would be challenging and likely inaccurate. **We will also add these arrows to the melt panels.**

---

## Author Response (AR1)

**Responses to Reviewer Comment**

Comments from the reviewer are given in black.

Author responses are in red, and amendments to the manuscript are given in bold red.

**RC1 - 'Comment on egusphere-2025-267'**

This study uses averaging of Cryosat-2 data to produce an 11-year record of centered-in-time annual DEMs for Pine Island Glacier, and then uses these to investigate the evolution of basal channels, primarily through a Lagrangian change analysis. Overall, I find the methods to be sound and the conclusions to be well-supported. I was particularly interested in the relationship between the pinning point and channel evolution. I think this will be a valuable addition to the literature on DEM methodologies, basal channels, and PIG.

We would like to thank the reviewer for their detailed comments and express our appreciation for the deep knowledge and expertise they brought to the review. As a result of these comments, the manuscript has been greatly improved.

If I've understood correctly, anomalies in elevation and basal melt rate are calculated by using a 7 km Gaussian filter, and then subtracting the results from the unsmoothed values. So, this is a neighborhood operation, and these anomalies are saying whether the ice base and melt rate are higher or lower than nearby points. For melt rates, as is described in lines 239-240, "A positive ice loss anomaly means more thinning and a negative ice loss anomaly shows anomalous thickening." It is *technically* correct to use this language; a negative ice-loss anomaly means thickening in the same way that a slightly weaker northerly wind is a southerly wind anomaly. However, the language is misleading, because based on the map in Figure 5, and from what we know about the PIG system, there should be no freeze-on occurring. So, in fact, there is no actual thickening due to melt (freeze-on). The anomaly values themselves are also not particularly meaningful, since they are comparisons to a fairly arbitrary section of ice-shelf base.

What I think *can* be said based on the anomalies is that, first of all, they are really only relevant if you're interested in the relative evolution of a distinctive basal feature such as a basal channel, and that a negative anomaly within the channel indicates that the amplitude of the channel is decreasing, i.e. the channel is filling in. That's interesting and important, but is currently obscured by the presentation in the manuscript.

Many thanks for the comments made here. Melt anomalies identify melt-induced, channel-scale thinning and thickening with respect to the larger scale. As pointed out by the reviewer, this does not mean freeze-on and we would not expect to see freezing anywhere on PIG. We agree that the clarity of the manuscript has been improved by carefully considering the wording used. In the example given by the reviewer, we have clarified our meaning by adding the following sentence on lines 272-274 in the revise manuscript: 'Therefore, if the focus is on a channel, a negative anomaly indicates the channel amplitude is decreasing and a positive anomaly indicates the channel amplitude is decreasing.'.

I think this can probably be addressed by being more careful with language within the current structure of the manuscript. However, I also think that a careful look at the results presented will reveal that not all of the discussion is relevant to the conclusions. Ideally, I think the results could be shortened quite a bit after reconsidering the meaning of the results and linking them to the conclusions.

The results section has been shortened by the removal of points not directly relevant to the results.

The length scale over which ice-flow divergence is calculated should be explicitly stated and justified.

Thanks for highlighting that this is missing from the manuscript. We have added the following sentence to the manuscript between lines 164 and 166: 'The velocity divergence, ∇·u, is calculated over a 480 m length scale (2 times the spatial resolution of the velocity dataset) using a centred calculation. For a given grid cell, divergence is calculated by the velocity gradient between its neighbouring cells. The output is therefore spatially centred on the initial grid cell of interest.'

In general, there is little to no treatment of error. While I don't know that a rigorous treatment of error is necessary, some discussion at least seems warranted. For example, calculations are made right up to the 2011 grounding line. It is not established how the grounding line has evolved after 2011 and throughout the record developed here. If it has retreated substantially, then using the 2011 grounding line is conservative and probably okay. If it has retreated slowly or not at all, then the measurements close to the grounding line are likely substantially influenced by not being fully in flotation, and these need to be discussed in the results and/or removed from the analysis.

Thanks for highlighting the need for a discussion around the treatment of error within the manuscript. Throughout the manuscript, we use a cautious 2011 grounding line for all analysis and datasets. The grounding line has retreated by a few kilometres during this period. While it remains that data near the grounding line is not fully floating, and therefore the assumptions of the methodology don't hold, all basal channel analysis supporting our conclusions is complete on areas of the ice shelf over 2 ice thicknesses downstream from the 2011 grounding line and therefore in an area of the ice shelf we consider to be fully floating. There are of course other types of error within this study. Lots of these are related to the CryoSat-2 surface elevation measurements. To address these issues we have added two paragraphs to the discussion between lines 395 - 413:

'We acknowledge several sources of error in both our methodology and data analysis. Noise in the CryoSat-2 data exists from it's poorly constrained firn penetration depth, errors inherent to the unwrapping processes and from orbit, range and angle uncertainties. The median standard deviation in a single DEM bin is 5.8 m. There are also errors within the velocity product used. Although we have not completed a formal error propagation here, we believe these uncertainties do not alter our conclusions but highlight the need for further investigation into how the datasets used impact ice shelf basal melt rate estimations.

Further errors arise from methodological assumptions. Most prominently, the method assumes the ice shelf is in hydrostatic equilibrium. We know this does not hold near the grounding line of an ice shelf. Throughout the analysis we have used a conservative 2011 grounding line to mask the grounded ice. All basal channel analyses supporting our conclusions have been completed in areas of the ice shelf further than 2 ice thicknesses from the 2011 grounding line and therefore in an area of the ice shelf we consider to be, on the large scale, fully floating. However, hydrostatic errors also exist on a basal channel scale. Observations have shown that the depth of the ice base can be underestimated within basal channel (Dutrieux et al., 2013; Rignot et al., 2025) due to bridging stresses within the ice. It has also been shown that the largest errors in deriving melt rates from remote sensing data occur along channel walls, where bridging stresses across ice-thickness gradients cause the ice shelf to deviate from hydrostatic equilibrium (Drews, 2015; Wearing et al., 2021). It is expected that melt is underestimated at the channel apex as a result (Wearing et al., 2021). This error is reduced as the ice relaxes as it moves away from the grounding line. This signal of relaxation is therefore aliased within our melt

observations. While such errors may affect the quantitative melt rates derived, we do not expect them to influence our qualitative conclusions, for example regarding Coriolis-favoured melting, as the errors should at least initially be symmetric across the channel.'

I'm also concerned that there appears to be no temporal interpolation of velocity fields during point migration – rather, it sounds like the velocity of a migrated point is pulled directly from the annual ITS\_LIVE velocity grid for the year into which the point fits. On an accelerating ice shelf, this can introduce large errors in migration location, which should probably be quantified. If the ice shelf is accelerating fairly steadily, linear interpolation between velocity grids would reasonably solve this issue. If I have misunderstood the method, it would be helpful to clarify the text.

Thanks for this methodological suggestion. The integration of velocity variability is something we have considered in great detail and is an involved part of our methodology. The reviewer has understood correctly that the velocity of a single point is pulled directly from the annual dataset corresponding to the year the point is in and is not interpolated in time using annual mean fields. For all point migration, we recalculate the velocity of the point on a monthly time step to account for changes in flow direction and speed as the point moves down the ice shelf.

On the creation of the manuscript, we made a conscious decision not to interpolate velocity fields in time. This decision was pushed by our goal to study/quantify basal channels and how they change in time, and we surmised that the velocity and velocity divergence both spatially vary on channel-sized (sub-kilometre) scales. Often, these changes are induced by the presence of a basal channel. Temporal interpolation between annual velocity estimates would therefore alias channelised velocity patterns. This will be especially prominent in cases where basal channels are not aligned with flow.

We have included a paragraph justifying our choices of velocity used in the study between lines 154 and 163:

'Throughout the method we use an annual velocity dataset. Over our observational period PIG accelerates and the flow bends west-ward (Figure A2) following the large calving events in 2019. Therefore using a time averaged velocity field would not be appropriate. However, using annual velocity datasets introduces discrete jumps at annual boundaries. During the advection process (both for time-centering and melt advection), the velocity dataset corresponding to the year of each time step is used. Similarly, the divergence field for the melt calculation is taken from the year of the target month. We decided against interpolating between velocity datasets to avoid smoothing the channel-scale features we aim to preserve within the velocity and divergence fields (Figure 6d). The aliasing of channel-scale features when interpolating between these fields would become more prominent when channels are not aligned along flow. Our choice ensures consistency throughout the method by aligning the treatment of velocity and divergence fields.'

"A dashed line is also plotted in Figure 4c and d. These correspond to the depth of the smoothed ice base in 2011 and 2017, respectively, and represent the ice base if channels did not exist." I'm not convinced of this equivalency – it's just a smoothed ice base, not necessarily one devoid of influence from channels.

Thanks for highlighting the ambiguity in our language here. Of course, this approach to creating a smoothed ice base and one void of basal channels is simplistic. The smoothed geometry is what we might expect if melt only resulted from ice-ocean interactions over a smooth ice base interface rather than one where the melt distribution is modulated by channel geometries. Of course, this is a simplistic assumption, as we would expect the channelised geometries to modulate melt efficiency. However, this analysis is merely trying to make the point that channel keels gave the ability to touch the seabed whilst a geometry without keels may not. **The sentence referred to now states: 'These correspond to the depth of the smoothed ice base in 2011 and 2017, respectively.' on line 235-236.**

The use of directions is confusing. The abstract uses east and west, lines 242-243 use north and south, and the description of Coriolis-influence melt uses northeast and southwest. This needs to be standardized. I realize that PIG is not directly N-S aligned, but since we normally think of ice as flowing from south to north in Antarctica, I'd personally stick to east and west.

**Thanks for highlighting this inconsistency. We have made this consistently East and West throughout the manuscript.**

"...and the surface elevation and melt anomaly are now in phase." I'm not sure what the point of this is, physically speaking. It's true that the graphs in Figure 7 go in the same direction at this point. But a negative melt anomaly means less melting, which would lead to relatively thicker ice, while a negative ice-thickness anomaly means thinner ice. So, physically, they are at odds with each other. While this is a mathematically correct statement, it doesn't seem to help much with the physical interpretation

We agree with the reviewer that this statement needs some clarification. Physically, the two lines being in phase means that, on average, there is less melting within the basal channel and more melting of the keels. In a steady state system, this means we would expect the channel to decrease in amplitude. This is interesting because it is the opposite of what is seen across the transect closer to the grounding line. On line 310-311, we have changed the sentence to:

'Despite the channel's consistent geometry only occurring in 2015, the time-averaged melt anomaly in this area remains strongly negative (Figure 7h).'

Line 281: "...where Channel 1 is carved, and a few kilometers downstream, where it is eroded." These terms don't make much physical sense. Carving is okay, but I assume the term "eroded" is supposed to mean where the channel is filling in (as the opposite of "carved"). Since erosion typically refers to taking material away, this would be a very confusing way to say it

Thanks for the comment. We have changed the above statement to say '...where the channels amplitude decreases through greater melting of its keels' to increase the clarity of this explanation. This is on lines 309-311.

Similarly, line 309 "indicating channel calving" – I assume carving was meant? Calving would not make physical sense here.

Thanks for spotting this mistake. This has been removed from the manuscript.

Figure 7 and the text show both Eulerian measures of change and a Lagrangian measure, but it is unclear what differing conclusions can or should be drawn from these differing analysis styles, and it is not brought into the discussion/conclusions. I do think there are some

interesting implications about the persistence of a channel imprinted on an ice parcel vs. the influence from Eulerian features such as plumes and pinning points that are hinted at in the manuscript, but could be stated much more explicitly.

Thanks for highlighting this. The Lagrangian measures allow us to track the evolution of a channelised section of ice throughout the observational period – gaining insights into the development of channels as they are advected downstream, their lateral movements and the associated melt rates. As pointed out, these discussions are somewhat different from those drawn from the Eulerian approaches, which instead focus on static sections of the ice shelf. We have emphasised these differences in the revised manuscript, for example, a discussion regarding channel wavenumber between lines 320 - 324: 'Initially spanning 2 km in width, Channel 1 persists throughout the observational period but undergoes geometric changes. Smaller channels, approximately 1–1.5 km wide, from the central, narrow feature. While the primary orientations of Channels 1 and 2 remain relatively stable, these smaller branches appear to veer westward, similar to the observations from Bindschadler et al. (2011). This process near doubles the channel wavenumber (the number of channels per unit distance) as the transect moves downstream (Figure 7i).'

I believe these features are discussed in some detail in Bindschadler et al. (2011), <a href="https://doi.org/10.3189/002214311797409802">https://doi.org/10.3189/002214311797409802</a>.

**This citation has been included on line 322.**

Figure 8 shows a few spots where the algorithm has marked the western flank as being shallower than the apex. I suggest that the algorithm is not working well in these cases, and perhaps these points should be excluded.

Thanks for highlighting this. We have added a criterion into our algorithm to stop this happening and reprocessed the results. The details of this criteria can be found of lines 345-346: 'The algorithm contains some criteria which the data must meet to be included in our analysis. Firstly, the depth of the channel apex must be shallower than that of the keels...'

Sergienko (2013) is a modeling study that clearly shows deflection of channels from flowlines.

Thanks for pointing this out. This point has been moved to the discussion in the new manuscript and we have included the suggested citation which can be found on line 446-447: 'This is contrary to suggestions from previous observations (Alley et al., 2024) and

some modelling studies (Sergienko, 2013), but aligned with other modelling studies (Gladish et al., 2012).'

West Antarctica abbreviated as WA: Since this is only used a small number of times in the first paragraph of the intro, it seems better to just write out "West Antarctica"

**This abbreviation has been removed throughout the manuscript.**

"data" should be plural throughout – e.g. line 86: "The 12 months of data that *are* required to *create* a single DEM *are* centred around..." (note also that "created" in this line should be "create"

**Thanks for pointing out this grammatical error. We have changed this throughout the manuscript.**

Line 195: "transects" should be "transect"

**This has been updated in the manuscript.**

Compound adjectives or nouns used as adjectives should technically be hyphenated – e.g. "kilometre-scale gradients" in line 223. Even terms like "sea-level rise" and "ice-shelf stability" should technically be hyphenated, although I realize that this isn't necessarily in style. If two nouns are not used as adjectives, they should not be hyphenated – e.g. remove hyphens from "ice-shelf" in lines 206 and 207

**We have updated this throughout the manuscript.**

Please replace rainbow color maps, which introduce false perceptions of high gradients, with a visually consistent color scheme

All rainbow colour maps have been removed from the manuscript. There are new colour maps in Figures 1,2,3,4,5,9 in the manuscript. These can be seen in Figures R1-5,9, respectively, at the bottom of this document.

In general, it's helpful if figure captions stand mostly on their own, so that a reader glancing through the paper can make sense of what you're doing. To that end, it would be helpful to define acronyms within captions (e.g. DROT in figure 1), and although it's fair to say that basemap and grounding lines are the same as in figure 1, it's a pain to go back through to find figure 1 to figure out when the basemap and grounding lines are from. Perhaps consider a legend within the figure for the grounding lines at least, and/or put the date of the basemap in the captions.

Thanks for pointing this out. We have removed the acronyms from the Figure 1 caption. The base map used throughout the paper is the Landsat Image Mosaic of Antarctica, and therefore doesn't have a date. We have kept it so these details are only in Figure 1 to avoid repetition.

Figure 3: The color bar appears to be very saturated, which limits the amount of information we can get from the figure.

We have extended the colour bar in Figure 3 to be +-20m instead of +-12m (see Figure R3 in this document).

Figure 4 (discussed in text): I wouldn't call those lines light blue, blue, and dark blue – at very least the first line is much more green than blue.

**We have changed this to refer to the lines as cyan, blue and dark blue on line 228.**

Figure 5: Titling this figure "The main variables needed to calculate basal melt rates" is odd when one of those variables is basal melt rates themselves. Although using "ice loss" as the color bar title is technically correct, it's a little confusing when on one color map it refers to basal melt (ice is completely lost and turned into water) and on another it's divergence (ice is

lost from the pixel but just moves next-door). Consider "ice-thickness loss" instead, here and in the text.

Thanks for highlighting the need for more clarity here. We have changed the colour bar label in Figure 5 to 'ice-thickness loss' and removed the titling 'the main variables needed to calculate basal melt rates'. See Figure R5 at the bottom of this document.

Figure 5 (described in text): "basal channels are clearly present within the thickness map." It would be helpful to mark them.

We have added the same cyan and magenta arrows to Figure 5 as are in Figure 6. See Figure R5 at the bottom of this document.

**Figure 7**

First, I'm not sure why the time-averaged elevation anomalies (and please clarify in the caption that these are anomalies) are plotted twice; I think it would either be better to plot them once on the same graph with the melt, or to plot the melt and elevation on separate graphs, but I spent a while trying to figure out why those were different.

Please also move the legend so it does not cover up some of those data.

Consider moving those averages to the right-hand side, since they're calculated from the Hovmöller data (just seems more logical, but this is a minor preference).

The caption statement "(b), (d), and (f) are overlaid with the zero contour line from (a), (c), and (e), respectively" has me confused. The zero contour line should come straight out of the Hovmöller, without need for averaging, so I'm not sure why it's linked to the averages. There's also what appears to be a zero contour line on the rest of the Hovmöllers, which aren't mentioned.

It would be easier to spot your cyan and pink arrows marking the basal channels if they were always on one side – probably on the right, since that's where the channels are consistently seen. It would also be helpful if the same locations were marked on the melt diagrams in d, h, and l.

We are grateful for the detail in which the reviewer has considered Figure 7. The time-average elevation anomalies are plotted twice to help guide the reader. Having them plotted alone alongside the elevation anomaly hovmoller first presents what these look like, and then keeping them on the second plots alongside the time average melt anomalies allows the reader to interpret the melt anomaly with respect to the elevation anomaly (interpret the melt anomalies within a basal channel context).

**We have moved the time-averaged plots to the right hand-side.**

We apologise for the mistake in the figure caption regarding contour lines and thank the review for noticing this. Following changes to the figure, the caption now reads: '(c), (g) and (k) are overlaid with the zero contour line from (a), (e) and (i), respectively, in black'

We have moved the channel markers to the right hand-side of the Eulerian plots but kept them on left hand-side on the Lagrangian plot. We have decided to keep the markers on the left for the Lagrangian plot because the channels evolve and split up by the time they reach the

right-hand side of this plot and therefore marking a single location would be challenging and likely inaccurate. **The channel markers have also been added to the melt panels.**

All changes to this Figure can be seen in Figure R7 at the bottom of this document.

**RC2 - 'Comment on egusphere-2025-267', Viet Helm**

In this study a new approach to calculate ice shelf basal melt rates on monthly resolution for Pine Island Glacier is presented. The study is a case study to demonstrate the new approach which uses CryoSat-2 swath data to derive monthly melt maps with a 250 m spatial resolution. The results show, that near the grounding line basal melting is 50% stronger on the western flank of a channel that on the eastern flank. In addition, findings suggest that the channelized geometries on PIG are triggered upstream and that the channel geometry facilitates ephemeral re-grounding as it moves downstream, potentially influencing ice-shelf stability. In general, the paper is well written, clearly structured and the presented idea to derive melt rates is novel and worth to be published in TC. Equations used are correct and figures are of good quality while mostly supporting the analysis.

We would like to thank the reviewer for their detailed comments on the manuscript and the expertise they have brought to the review. Following their comments, the manuscript has been greatly improved.

While reading I think that the results part is very voluminous and already includes sections which should be more presented in the discussion section (e.g. L202 – L225). Please check carefully.

Thanks to the reviewer for highlighting this – we have distilled the results section and focused text in the discussion where appropriate. For example, the discussion regarding Coriolis favoured melting has been moved to between lines 436 and 444 in the Discussion section: 'Our findings lend strong support to theories positing that channelised melt is concentrated on the Coriolis-favoured flank, where rising buoyant plumes are deflected by Earth's rotation (Gladish et al., 2012; Millgate et al., 2013; Marsh et al., 2016; Gourmelen et al., 2017; Cheng et al., 2024). Within 15 km of the grounding line, we observe ocean driven melting to be approximately 50% higher on the western flank than the eastern flank, with the highest melt rates on the upper portion of the western flank (Figure 8). Higher melt rates are also evident on the upper portion of the eastern flank relative to its lower portion. Our results are consistent with other studies positing enhanced melting on the channel apex. Notably, this pattern reverses downstream, where the eastern keel exhibits the highest melt rates. This can be explained by the deeper eastern keel being exposed to warmer (deeper) waters and at a lower pressure-dependent freezing point. Alternatively, mid-water intrusions arising from the calving front and an associated enhanced circulation could focus melt on this flank instead.'

Shean (2019) used high resolution Tandem-X DEMs to derive melt rates of PIG. They found a higher melting associated with basal channels and deep keels near the grounding line and relatively shallow keels over the outer shelf and do not discuss a more pronounced western flank melting. It would be nice and important to see if you can confirm and discuss the findings

of Shean in more detail and also if this Coriolis dependent melting as it was not discussed in Shean (2019) might be a result your processing methodology.

Thanks for highlighting the need for more discussion on this point. We are only able to deduce the Coriolis-favoured melt pattern because of the large number of observations within our new dataset. Compounding and averaging these observations allows us to detect such a signal over a relatively small spatial scale within relatively noisy data. Shean (2019) didn't comment on Coriolis-dependent melting. Their dataset only contained a single melt map, which spanned 2008-2015 and was calculated by averaging all melt rates that intersect a given pixel in the time period. It is therefore likely that either they didn't have enough data to detect this Coriolis signal, or this signal was aliased by the composite method used. Nevertheless, it is interesting that this signal was not reported within their dataset. We have added a paragraph (lines 421-435) in the discussion comparing our results to those of Shean et al., (2019): 'Our observations show that melt rates near the grounding line are concentrated within basal channels (Figure 8). While these observations agree with many others (Dutrieux et al., 2013; Marsh et al., 2016; Stanton et al., 2013; Gourmelen et al., 2017; Humbert et al., 2022; Zinck et al., 2023), it also contrasts observations made on PIG ice shelf by Shean et al. (2019). There are a number of reasons why we might see these differences in our observations. Firstly, Shean et al. (2019) only noted the highest melting on channel keels within 3-4 km of the grounding line, an area where the hydrostatic assumption begins to breakdown. Further, in their study, channels were defined as any areas where the surface elevation anomaly was <-1 m (equivalent to a thickness anomaly of~9 m), and keels where the elevation anomaly was >1 m (thickness anomaly greater than~9 m). It therefore likely that observations classified as a keel in Shean et al. (2019) are defined as channel flank in this study and the discrepancy arises from different labelling choices. We also note that the algorithm used in Shean et al. (2019) identifies large portions of the grounding zone to be basal keels which might also affect the results. Furthermore, their channelised melt conclusions were deduced from a single composite melt map that spanned 7 years. It is therefore likely that across channel melt variations were smoothed and hence no conclusions were drawn regarding Coriolis favoured melting. Conversely, our conclusions derive from analysis over only 2 persistent and larger channels, and a broader, more systematic study bringing in more observations (and associated caveats about the representations of smaller scale features) may provide a more nuanced picture.'

As a general remark. How much is the velocity changing throughout the observed time period? Can you please make a figure in the supplements showing the difference of the x and y components of each single velocity field to an averaged velocity. To my opinion it would be much better to use an averaged field in the whole processing as long as the velocity is more or less constant throughout the time period. This would substantially minimize errors in the Lagrangian shift, which is based on pixel to pixel shifts and therefore very sensitive to noise of the velocity components on the pixel scale. I think this is important, especially as you try to analyze small scale melt differences within a channel.

In the creation of this manuscript, we considered the methodology used in great detail. Figure R10 at the bottom of this document shows the difference in each annual velocity field from the time averaged field. The velocity has increased by ~1000m/yr during this time, and the flow diverted westward after 2018. Due to these changes, we don't think an averaged velocity field would be appropriate here. We have added a supplementary figure (Figure A2) showing the velocity change with respect to the time averaged velocity and we have included a

comment on this on line 154-156: 'Over the observational period PIG accelerates and the flow bends west-ward (Figure A2) following the large calving events in 2019. Therefore, using an averaged velocity field would not be appropriate.'

Suggestion for all overview figures: I think the western part of the ice shelf doesn't need to be included as it is never discussed in the paper. Please zoom in to the main fast flowing part which you are focusing on (Fig 1, Fig2, Fig3, Fig4b, inlet Fig9, Fig A1, A2)

Thanks for your comments on the figure layout. While the western part of the ice shelf isn't directly discussed in the manuscript, the area has been highly dynamic during this period and significant changes have been observed. We therefore argue that keeping this in the figures is both useful for orienting the reader and of interest to some readers.

In addition, the choice of the selected along and across flow segments which are varying in the paper make it difficult for the reader to follow. Can you please include in your overview Fig1 also the cross section X-X', Y-Y', Z-Z' and the box S-S' and U-U'

This was something we considered in the creation of the manuscript. We acknowledge that the number of different sections and transects is a little involved and requires the reader to take special care. However, having all transects on one figure is messy and harder for the reader to decipher. We have tried to place the relevant transects near where they are used in the manuscript to aid the reader. For example, the transects used in both Figures 4 and 9 are displayed within the Figures.

Z-Z' seems to be very similar to B-B' – why not using the same.

Z-Z' was chosen specifically to cross the area of ephemeral grounding in 2017, whereas B-B' is a more general location downstream of the initial 2011 pinning point and away from the highest melt rates near the grounding line. We have chosen to keep them as separate lines as there would be no real benefit to changing them. Z-Z' needs to be displayed on Figure 4 for clarity. If we made B-B' Z-Z', that would be asking readers to check two different Figures to find the location of the Hovmoller transects and comparing their locations on the ice shelf would be more challenging as a result.

Can you please slightly enlarge S-S' that it includes Y-Y', Z-Z' and B-B' and please also mark in Figure 5 and 6 the position of those cross sections. This would help to understand better otherwise it's a bit confusing.

Thanks for highlighting this. We have enlarged the area of S-S' in Figures 5 and 6 to include the transect areas. However, we haven't marked the transects on these figures as we found it looks confusing for the reader.

Please don't use rainbow color scale for elevation and thickness.

All rainbow colour maps have been removed from the manuscript. There are new colour maps in Figures 1,2,3,4,5,9, which can be seen at the bottom of this document in Figures R1-5,9, respectively.

here you claim that the new method can be applied on every ice shelf or terrain. I think this is not correct. You still need sufficient data coverage of the swath data to form this DEMs based on

1 year of acquisition data. And this is not the case for most of the ice shelves. For PIG or Dotson it seems to work.

Thanks for highlighting this. We have added a caveat to line 72-73 which says '...this method can be applied to any ice shelf with sufficient surface elevation altimetry data'. We have also commented on the need for an across-track slope for the swath processing, which isn't always the case. Lines 87-91 state 'We also note that the swath processing relies on the presence of an across-track surface slope to prevent left-right phase ambiguity. This can be challenging over some ice shelves. However, many studies have shown that we can retrieve swath elevation measurements for a significant portion of CS2 waveforms over many ice shelves '(Gourmelen et al., 2017; Davison et al., 2023; Surawy-Stepney et al., 2023; Wuite et al., 2019; Goldberg et al., 2019)'

Usually from theory quasi nadir swath processing is affected by phase ambiguities and low coherence over flat terrain, where across track slope is less than half the antenna beamwidth – which usually is the case over ice shelves. As Cryosat2 has a small mis pointing of 0.1° this left and right looking phase returns are not completely canceling, allowing to detect some coherence and therefore to derive elevation estimates across track in some places like PIG. I think it is important to be mentioned.

Thanks for raising this point. The swath processing does rely on the presence of a surface slope to be viable, and for very low slopes, a left-right ambiguity can occur and usually results in reduced signal quality. This has been addressed in the response to reviewers in Gourmelen et al., 2025. While a number of previous studies have shown we can retrieve swath elevation for a significant portion of CryoSat-2 waveforms over ice shelves (Gourmelen et al., 2017; Wuite et al., 2019; Lilien et al., 2019; Goldberg et al., 2019; Wei et al., 2020; Davison et al., 2023; Gwyther et al., 2023; Surawy-Stepney et al.,, 2023), we hypothesize that it has to do with ice shelf heterogeneity in backscattering properties across ice shelves, and to the fact that CryoSat-2 is not pointing perfectly at nadir but is mis-pointing slightly (Recchia et al., 2017). Following on from the response to the above comment we go on to comment on the contribution of the CryoSat-2 mis-pointing in lines 91-92: 'It is thought that the slight mispointing of CryoSat-2 prevents the complete cancelling of returns and therefore contributes to the coherence of data in these regions (Recchia et al., 2017).'

Here you argue that the time centring method is more accurate than just binning and compare both DEMs in A1. How do you know which one is better? Of course you reduce the averaging of across flow features but with the time centering method you also introduce errors which are related to the velocity field and it's derivative, which I assume is very noisy when using yearly velocity fields (see comment above). I would suggest to compare to other high resolution DEMs like the Tandem-X DEM presented in Shean (2019) to evaluate which method is better.

Thanks for raising this point. We have completed a comparison with one of the DEMs presented in Shean et al., (2019) to show that time-centring is more accurate. Figure 1 (Figure R1 below) now also shows that the spatial distribution of channels in the time-centred approach agrees with that of Shean et al., 2019 (Figure 1b/R1b). It also now shows that the time-centred DEMs correlate better with the Shean DEMs than the direct method (Figure 1c/R1c), particularly for the across-flow channels. Further, we have updated Figure A1 to further highlight that the time-centring method better captures the spatial distribution than the direct method with respect to the Shean et al (2019) DEM. This is discussed in the new manuscript between lines 115 and 126:

'By adopting the time-centring process, we reduce the aliasing of these features (Figure 1c). Along the main trunk of the ice shelf the distribution of surface troughs and ridges (associated with basal channels and keels) in our time-centred DEMs show good agreement with a higher-resolution, but less frequently acquired, WorldView-derived DEM from Shean et al. (2019) (Figure 1b). While our CryoSat-2 DEMs successfully capture the primary longitudinal channel features, they do not resolve small transverse channels as clearly. However, the time-centring method captures these surface features with substantially greater accuracy than the direct method (Figure 1c). Between C' and C" of the transect shown in Figure 1b, the correlation with the Shean et al. (2019) DEM improves from -0.05 using the direct method to 0.63 with the time-centring approach.

Despite the overall consistency in feature distribution, our CryoSat-2 DEMs underestimate the amplitude of channels relative to the Shean et al. (2019) product (Figure 1c). This underestimation occurs primarily through an underestimation of the surface ridges. While there are a number of reasons for this bias (firn penetration, averaging in time etc), we assume that it is consistent in time and therefore does not impact the temporal change observed throughout this paper.'

I would also like to see a typical point cloud coverage of one month of swath elevations to get an idea of the general data coverage.

See figure R11 at the bottom of this document for an example of a month's worth of point cloud coverage.

As you use the median as a very robust averaging method, I would like to see the standard deviation for each pixel to get an idea of how much the swath elevation point cloud elevations are varying within one pixel.

See Figure R12 at the bottom of this document for an example of the standard deviation for our 2018/06 DEM.

Do you filter or exclude outliers of the swath data before averaging?

Thanks for raising this. We do quality check the data before creating the DEMs. We have included details between lines 92-94: 'Once the corrections have been applied, any data of low quality has been removed. If the return has a coherence of less than 0.6, or it's absolute difference with the Reference Elevation Model of Antarctica (REMA) is greater than 100 m, then the data has been removed.'

Can you please exactly explain where H is coming from? Due you use the mean(h) of both contributing DEMs, or h of the first DEM or a constant h for all time steps?

We have added the following on line 146: 'H is taken from the initial DEM.'

Is the noise of the monthly melt maps maybe related to the noisy velocity field?

It is possible that the noisy velocity fields are contributing to the noise in the monthly melt maps. It is also likely that some of the melt map noise is coming from noise in the CryoSat-2 DEMs. Despite this, we think that remaining with the slightly noisier velocity fields is more appropriate that using a 'cleaner' time-averaged product that doesn't capture the evolution of the ice shelf during this time (**Figure A1/R10** in this document). We have included a sentence

in the manuscript between lines 154-157 to address this: 'Throughout the method we use an annual velocity dataset. Over our observational period PIG accelerates and the flow bends west-ward (Figure A2) following a large calving event in 2018. Therefore, using a time averaged velocity field would not be appropriate. However, using annual velocity datasets introduces discrete jumps at annual boundaries. Furthermore, the annual velocity datasets are relatively noisy and this likely contributes to some noise in our melt maps.'

The averaging of monthly data to reduce noise is a well know procedure. However, I do not understand why you advect the melt maps. To my opinion this is not correct. The overall melt pattern is not moving in a Lagrangian sense with the ice. The melt is triggered by ocean water masses and the ice is moving. This means that the melt pattern can locally change with time due to changes in warm water supply through the ocean but this warm water supply is decoupled from the ice movement. Therefore, an averaging of monthly melt maps should be done without advection. I also think because of this additional advection you change the melt distribution across a channel. And this change is correlated to the across flow component of the velocity field. Therefore, I would like to recommend to redo the analyses based on not advected averaging and see new figures 6,7,8. Will this change your conclusion of pronounced western flank melting and support the findings in Shean of high keel and channel melting in areas close to the grounding line?

As mentioned, please redo the analysis with not advected averaged melt maps.

Thanks for your suggestion on this. While melt is triggered by ocean water masses delivering heat to the ice-ocean interface, this process is not decoupled from ice movement. The ability of the ocean to deliver this heat to the ice-ocean interface depends on the cavity circulation, which is largely modulated ice geometry, particularly in a plume context within a basal channel. Therefore the ice movement matters for ocean currents and therefore also the spatial distribution of basal melt. A temporal averaging that doesn't take this into account would smooth any channelised melt anomaly that is not aligned with flow. To check our understanding, we computed melt ratees with Eulerian averaged melt maps. Figure R10 at the bottom of this document shows the Hovmoller (Figure 7 in the manuscript) with Eulerian averaged melt maps. The melt signal is heavily smoothed across all 3 melt panels. While the conclusions here remain the same as the Lagrangian averaging method, this is primarily because we are focusing on two along flow channels. If we were looking at channels that aren't directly along flow, which will hopefully be possible in the future, the Eulerian method would smooth over the melt signal within them. We are therefore choosing to continue with the Lagrangian averaging method because it is the most generalisable method that would preserve across-flow melt signals.

We have added a paragraph justifying our choice between lines 174 – 178: 'The averaging of annual melt maps was complete within a Lagrangian framework to avoid smoothing channelised melt anomalies that are not aligned with flow. While ocean conditions set the amount of heat in the cavity, the melting caused depends on the ability of ocean circulation to bring the heat to the ice-ocean interface. These currents depends on ice shelf geometry on both large scales and channel scales. Therefore, ice movement changes the spatial distribution of melt and a Eulerian averaging would smooth over these changes.'

Please also give the equation of how you derive ice shelf base showing in Figure 4.

This equation has been included as Equation 4 in the paper, between lines 226 and 227:

$$d = -h(\frac{\rho i}{\rho w - \rho i}).$$

This is an interesting finding, that you are able to see an un- and regrounding in the CS2 data. Can you please confirm that DROT also see an ungrounding of section Z-Z' in 2023?

Yes, the DROT data also shows an ungrounding in 2023. We have included this on line 230-231 in the new manuscript: '...when the DROT method detects no further grounding and the longer term impact of the unpinning on the distribution of basal channels can be assessed.'

In Fig 4b you only show DROT grounding of 2017. I would suggest to zoom in in Fig 4B to only show the relevant ice shelf section and enlarge labelling in 4B. Maye provide another figure like 4B with DROT grounded areas of 2023 to further support the CS2 data.

Thanks for highlighting this. As mentioned above, the DROT method doesn't detect any grounded ice in 2023. We have decided to keep the map as it is to provide a broader context for the transects.

As suggested. Why not use a constant averaged less noisy velocity field to avoid effects of changing and noisy divergence.

Please refer to the response above and Figure A1/R10 regarding the change in velocity over the observational period.

Could you please include in your analysis and Figure 8c the melt rate within the channel apex as well as the melt rates of the western and eastern keel. It would be interesting to see if this Coriolis effect, as proposed, is only dominant in the channel and if in the neighboring keels a different melt rate is observed. Furthermore, this would also show whether Shean's observations of higher melt rates at the Keels can be confirmed.

Thanks for this comment. In response to this comment, we have calculated the melt rates within the channel and keel on the west and east of the channel i.e. splitting the channel into 4 transverse sections as opposed to the 2 (west/east) in the current manuscript. The results show a gradient between the 4 sections, with the highest melt near the grounding line remaining on the western keel, followed by the western apex, eastern apex and then eastern keel. Figure 8 and the surrounding text has been updated to include this new analysis. The new Figure 8 can be seen in Figure R8 at the bottom of this document.

These results still don't support Shean et al (2019) observations of highest melt rates on the keels. However, Shean et al (2019) defined a keel to be where the ice shelf surface elevation anomaly is greater than 1m. This corresponds to a thickness anomaly of ~10m and therefore a base depth anomaly of ~9m. It is therefore likely that, through this definition, melt rates on the channel flank are being labelled as channel keel. We have added a detailed paragraph (lines 421-435) in the discussion about the discrepancies between our results and those of Shean's regarding this point: 'Our observations show that melt rates near the grounding line are concentrated within basal channels (Figure 8). While these observations agree with many others (Dutrieux et al., 2013; Marsh et al., 2016; Stanton et al., 2013; Gourmelen et al., 2017; Humbert et al., 2022; Zinck et al., 2023), it also contrasts observations made on PIG ice shelf by Shean et al. (2019). There are a number of reasons why we might see these differences in our observations. Firstly, Shean et al.

(2019) only noted the highest melting on channel keels within 3-4 km of the grounding line, an area where the hydrostatic assumption begins to breakdown. Further, in their study, channels were defined as any areas where the surface elevation anomaly was <−1 m (equivalent to a thickness anomaly of∼9 m), and keels where the elevation anomaly was >1 m (thickness anomaly greater than∼9 m). It therefore likely that observations classified as a keel in Shean et al. (2019) are defined as channel flank in this study and the discrepancy arises from different labelling choices. We also note that the algorithm used in Shean et al. (2019) identifies large portions of the grounding zone to be basal keels which might also affect the results. Furthermore, their channelised melt conclusions were deduced from a single composite melt map that spanned 7 years. It is therefore likely that across channel melt variations were smoothed and hence no conclusions were drawn regarding Coriolis favoured melting. Conversely, our conclusions derive from analysis over only 2 persistent and larger channels, and a broader, more systematic study bringing in more observations (and associated caveats about the representations of smaller scale features) may provide a more nuanced picture.'

Can you please mark with a grey bar the position and extend of the channel you discuss in L379 ff

We have marked Channel 1 on Figure 9 with a pink arrow to maintain consistency with the other figures – See Figure R9 below.

Figure R1: (a) A Digital Elevation Model (DEM) of Pine Island Glacier from 2017 derived from CryoSat-2, with transects corresponding to sections used in Figure 7 and (c). In (a) and (b) the 1996 and 2011 MEaSUREs grounding lines (Rignot et al., 2016) are shown in black and magenta, respectively. (a) is also overlaid with areas of ephemeral re-grounding in 2017 as observed from Sentinel-1 using the differential range offset tracking method (described in Section 2.3) in cyan. The background in (a) and (b) is a greyscale Landsat Image Mosiac of Antarctica (LIMA) 15m. (b) shows a high-pass filtered CryoSat-2 DEM from December 2012. It is overlaid with the 0 m contour line from Shean et al. (2019) DEM in black. (c) Shows elevation profiles from our time-centered DEMs, our DEMs using the direct method and from the Shean et al. (2019) DEMs along the transect between C, C' and C" shown in yellow on (b). The correlation of each of our methods with the Shean et al. (2019) DEM along the two sections of this transect are also shown.

Figure R2: (a) Digital Elevation Model derived from CryoSat-2, (b) velocity, coloured by velocity magnitude and overlaid with velocity vectors, (c) velocity divergence, and (d) basal melt rates. All components shown are in 2015. The black rectangle in (a) is for Figures 5 and 6. The basemap and grounding lines are as in Figure 1.

Figure R3: (a)-(I) show the annual elevation anomalies with respect to the first DEM in June 2011 for 2012-2023, respectively. Basemap and grounding lines as in Figure 1.

Figure R4: Depth of the ice shelf base in 2011, 2017 and 2021 across three different transects. (b) shows the three transects used in the other subplots. (a) shows the depth of the base between X and X', (c) shows the depth of the base between Y and Y', and (d) shows the depth of the base between Z and Z'. The red lines in (b) show the areas where Sentinel-1 DROT data indicates grounded ice in 2017. The vertical grey shaded areas in (a),(c) and (d) show grounded areas in 2011 and 2017 as observed by the MEaSUREs 2011 grounding line and DROT 2017. The black dashed vertical lines mark the points of intersection between transects. The brown shaded areas on (a), (c), and (d) are the seabed from BedMachine merged with observations from (Dutrieux et al., 2014b).

Figure R5: Thickness (a), Lagrangian thickness changes (b), basal melt (c) and ice flux divergence (d) in the black rectangle in Figure 2a.All variables are from 2015.

Figure R6: High-pass filtered anomaly of thickness (a), Lagrangian thickness change (c), basal melt (c) and ice flux divergence and velocity (d) within the black rectangle in figure 2a. All variables are from 2015. The pink and cyan arrows indicate Channels 1 and 2, respectively, as discussed in the text.

Figure R7: Figure 7. Elevation and melt anomaly evolution for different sections on the ice shelf. (a) and (c) are the elevation and melt anomalies between A and A' in Figure 1. (e) and (f) are the elevation and melt anomalies between B and B' in Figure 1. (i) and (k) are the elevation and melt anomalies originating from A and A' in Figure 1 in a Lagrangian framework. This transect has been advected forward in time until it is between a and a' at the end of 2020. (c), (g) and (k) are overlaid with the zero contour line from (a), (e) and (i), respectively, in black. The right hand column of panels shows the time averaged value of elevation and melt from the adjacent panel. The pink and cyan arrows indicate Channels 1 and 2, respectively, as discussed in the text.

Figure R8: Comparison of melt rates on different sections of Channel 1 as a function of distance from the 2011 grounding line.

Figure R9: The brown area is the ice bed upstream of the grounding line along the black transect on the inset map, as measured from airborne radar (Vaughan et al., 2012). Using ITS\_LIVE annual velocity data, the location of this flight line has been advected downstream, with the annual location of these advected transects shown on the inset map. The surface elevation and ice base depth derived from CryoSat-2 along each of these transects has then been plotted.

 $\label{eq:Figure R10:PIG} \textbf{Figure R10: PIG annual velocity difference with the time averaged velocity over this period.}$

Figure R11: Example of one month of CryoSat-2 data over PIG.

Figure R12: 2017/06 DEM Standard Deviation.

Figure R13: Hovmoller (figure 9) with Eulerian averaged melt maps.

---

## Referee Report (RR1)

Thanks for the really thorough revisions based on the previous comments from me and the other reviewer. I think the paper reads much more clearly now. I have a number of comments below, most of which are just grammar comments. A few are substantive to the science. The one that's probably most important is the idea put forward in the manuscript that the formation of channels makes deep keels that can keep an ice shelf pinned longer than otherwise. I had commented last time about viewing a smoothed ice base as representative of the ice shelf without channels, but I hadn't quite understood where that argument was intended to go in the manuscript. Now I think I understand the message, but I don't see any way this can be true. Channelized melt does indeed thin ice within the channels, leaving relatively thick keels behind. However, those keels aren't any thicker than they would be without channels, and in fact are thinner due to the secondary flow into the channels, which is documented here. So, channels can change the distribution of thick/thin ice, but they don't build keels that would help keep an ice shelf grounded longer than it would be without keels. I think that message needs to be refined. With that adjusted, and a couple other things mentioned below, I think this paper will be a really strong contribution to the literature.

Lines 93-94: "data... has" should be "data... have" (two instances)

Lines 94 and 165: "it's" should be "its"

Lines 164-166: The choice of length scale makes sense based on your grid size, but it really needs to be compared to the features of interest in order to be useful. Any sort of gradient calculation looks at a signal between points, and the farther the points are apart, the more that a local signal is diluted. I'd say that the scale chosen is fairly reasonable – your channels are all >1 km across, so a length scale of 480 m will capture signals within channels without smoothing the signal a lot due to sampling points outside the channel, or even into adjacent channels, although it will still spread the signal at least somewhat at the channel edges. This is what this section should state and justify.

Lines 174-178: Just a note that I strongly agree with the authors that the melt rates also need to be advected in a Lagrangian framework

Paragraph starting in line 215: Consider mentioning the 2017 re-grounding of the pinning point in this paragraph. It's thrown in a bit later like that's something the reader should already know, so it would be helpful to mention here when you're establishing the history of the pinning point.

Line 221: "Unground" should be "ungrounded"

Lines 251-254: "In both Figures a hypothetical smoothed version of the ice shelf base would not have grounded, instead they would have left a 30-meter deep cavity between the

ice shelf base and the seabed. This suggests that without a pronounced channel and keel geometry on the ice shelf base, the pinning point might not have been sustained for as long, and the thick ice may not have periodically re-grounded."

As noted in the intro to this review, I'm not buying this argument. Melting in basal channels thins the ice within the channel, but I haven't seen any evidence that the presence of basal channels inherently thickens keels, making them more likely to ground. On the contrary, secondary flow would suggest that convergence induced by the presence of a basal channel (due to ice-thickness gradients) would promote thinning of keels, although not matching the rate of thinning inside the basal channel due to melt. So basal channel formation should thin an ice shelf overall, while also adding a lot of complex basal topography. I don't see how an ice shelf without basal channels would then result in an ice-shelf base 30 m shallower than what is observed.

I certainly buy that basal channels can locally thin the ice, changing where it's thick enough to reground. It also makes complete sense that a pinning point could locally thicken the ice, and when released, cause grounding downstream when the thicker ice advects. So, it's fair to say that basal channels change the patterns of grounding and re-grounding. But I don't think it's fair to say that basal channels in themselves would keep anything grounded longer or promote re-grounding. They promote complexity, but overall thinning.

Lines 268-269: "In the area shown, basal channels are clearly present within the thickness map (Figure 5a, marked by the cyan and magenta arrows). However they cannot be seen in the thickness change variables (Figures 5b-d)." Maybe it's worth mentioning here that this isn't necessarily very surprising, and it doesn't indicate that there's something wrong with the measurements. The calculations are Lagrangian, so all the lack of channelized pattern in the dH/dt and flux divergence says is that the parcel in the downstream measurement hasn't changed a whole lot as compared to when it was at the upstream measurement, and that change is similar inside channels as outside channels. Or in this case, it just means that the amplitude of the channel isn't changing very much as you move downstream.

Line 313: "Channel's" should be "channels," and near the end of the line, "channels" should be "channel's"

Line 315: "depth-dependent" should be hyphenated

Line 321: "Smaller channels, approximately 1–1.5 km wide, from the central, narrow feature." This isn't a complete sentence.

Line 323: "near" should probably be "nearly"

Line 337: "across-channel" should be hyphenated

Line 396: "it's" should be "its"

Lines 398-399: "we believe these uncertainties do not alter our conclusions" probably needs to be backed up. Why do you believe this? Do you believe they're consistent in time? Or too small compared to the signal?

Line 431: "across-channel" should be hyphenated

Lines 445-447: "Despite this asymmetry in melt rates, the channel apex doesn't deviate from the flow lines when temporal variability of ice velocity is taken into account (Figure 7i). This is contrary to suggestions from previous observations (Alley et al., 2024) and some modelling studies (Sergienko, 2013)..."

Be very careful with this. The results from Alley et al. (2024) and Sergienko (2013) suggest that basal channels should deviate somewhat from flow lines in steady state – i.e., if you look at a plan-view image of a basal channel, and you look at plan-view flow lines that take into account the paths that ice parcels have taken throughout their journey across the ice flow (i.e. including history of ice-flow changes), the channel will be deflected slightly to the left of those flow lines when looking downstream. This isn't an analysis that you've done in this paper. The two Eulerian analyses in Figure 7 would only show leftward migration if you have a non-steady Coriolis-influenced melt signal, i.e. if the Coriolis-influenced melt at that location is increasing over time. If your channels are not melting more intensely over time, which appears to be consistent with your data, you would not expect to see leftward migration in these. Your Lagrangian analysis in Figure 7 should show leftward migration, but only where you have Coriolis-influenced melt on the left-hand side of the channel. Figure 8 suggests that this is only true near the head of the channel, and Channel 1 certainly shows somewhat of a curve in that direction in the upper reaches. If your melt rates are correct, this almost has to be true – if it's melting faster on that side, the apex of the channel will necessarily move in that direction as the ice advects downstream. Your melt-rate differences between flanks decrease or reverse downstream, so we'd expect to see that effect disappear or reverse (and I'm guessing it will be very small either way with those similar magnitudes downstream). Furthermore, the smaller branches you show downstream "veer westward" in the Lagrangian analysis, which would be consistent with Coriolis-favored melt in these features. I think this paragraph needs to be reworded, because in this context I don't think any of the papers cited and the analysis presented here are inconsistent with each other.

Line 452-457: As above, unless I'm missing something, I don't think this is supported. Basal channels don't cause thicker keels, they just determine where thick regions are left behind during melting.

---

## Author Response (AR2)

**Responses to Reviewer Comment**

Comments from the reviewer are given in black.

Author responses are in red, and amendments to the manuscript are given in bold red.

**RC1 – 'Comment on egusphere-2025-267'**

Thanks for the really thorough revisions based on the previous comments from me and the other reviewer. I think the paper reads much more clearly now. I have a number of comments below, most of which are just grammar comments. A few are substantive to the science. The one that's probably most important is the idea put forward in the manuscript that the formation of channels makes deep keels that can keep an ice shelf pinned longer than otherwise. I had commented last time about viewing a smoothed ice base as representative of the ice shelf without channels, but I hadn't quite understood where that argument was intended to go in the manuscript. Now I think I understand the message, but I don't see any way this can be true. Channelized melt does indeed thin ice within the channels, leaving relatively thick keels behind. However, those keels aren't any thicker than they would be without channels, and in fact are thinner due to the secondary flow into the channels, which is documented here. So, channels can change the distribution of thick/thin ice, but they don't build keels that would help keep an ice shelf grounded longer than it would be without keels. I think that message needs to be refined. With that adjusted, and a couple other things mentioned below, I think this paper will be a really strong contribution to the literature.

We would like to thank the reviewer for their time and expertise in giving more detailed and insightful comments. We have addressed the reviewers' comments regarding the formation of deep keels and their interactions with pinning points below.

Lines 93-94: "data... has" should be "data... have" (two instances)

Thanks for noticing this grammatical error. We have changed these to 'data... have' as suggested.

Lines 94 and 165: "it's" should be "its"

**We have changed this.**

Lines 164-166: The choice of length scale makes sense based on your grid size, but it really needs to be compared to the features of interest in order to be useful. Any sort of gradient calculation looks at a signal between points, and the farther the points are apart, the more that a local signal is diluted. I'd say that the scale chosen is fairly reasonable – your channels are all >1 km across, so a length scale of 480 m will capture signals within channels without smoothing the signal a lot due to sampling points outside the channel, or even into adjacent channels, although it will still spread the signal at least somewhat at the channel edges. This is what this section should state and justify.

Thanks for highlighting the need for a discussion on this in the manuscript. The following has been added to the end of the mentioned paragraph on lines 166-169: 'This scale is relatively small with respect to the channel width, so we expect to capture the channelised signal of divergence without smoothing the signal within a channel. However, it is likely that signals at the channel edge are somewhat smoothed as the calculation computes gradients between points on either side of the channel boundary.'

Lines 174-178: Just a note that I strongly agree with the authors that the melt rates also need to be advected in a Lagrangian framework.

**Thank you!**

Paragraph starting in line 215: Consider mentioning the 2017 re-grounding of the pinning point in this paragraph. It's thrown in a bit later like that's something the reader should already know, so it would be helpful to mention here when you're establishing the history of the pinning point.

Thanks for noticing this is missing in the paper. We have added the following sentence to the mentioned paragraph: 'The thick ice ephemerally reground as it advected downstream (Joughin et al., 2016)'.

Line 221: "Unground" should be "ungrounded"

**We have changed this.**

Lines 251-254: "In both Figures a hypothetical smoothed version of the ice shelf base would not have grounded, instead they would have left a 30-meter deep cavity between the ice shelf base and the seabed. This suggests that without a pronounced channel and keel geometry on the ice shelf base, the pinning point might not have been sustained for as long, and the thick ice may not have periodically regrounded."

As noted in the intro to this review, I'm not buying this argument. Melting in basal channels thins the ice within the channel, but I haven't seen any evidence that the presence of basal channels inherently thickens keels, making them more likely to ground. On the contrary, secondary flow would suggest that convergence induced by the presence of a basal channel (due to ice-thickness gradients) would promote thinning of keels, although not matching the rate of thinning inside the basal channel due to melt. So basal channel formation should thin an ice shelf overall, while also adding a lot of complex basal topography. I don't see how an ice shelf without basal channels would then result in an ice-shelf base 30 m shallower than what is observed.

I certainly buy that basal channels can locally thin the ice, changing where it's thick enough to reground. It also makes complete sense that a pinning point could locally thicken the ice, and when released, cause grounding downstream when the thicker ice advects. So, it's fair to say that basal channels change the patterns of grounding and re-grounding. But I don't think it's fair to say that basal channels in themselves would keep anything grounded longer or promote re-grounding. They promote complexity, but overall thinning.

Thanks to the reviewer for highlighting their concerns with our argument here and ensuring that we have made our opinion clear. We agree with the points made by the reviewer that, of course, channelised melting doesn't result in the growth of keels. However, we make the point from the perspective of otherwise uniform melting. If an ice shelf experiences X Gt/yr of melt, the deepest point of the ice draft will depend on where this melt is applied. If the melt is applied uniformly across the base of the ice shelf (i.e. what we might expect if the base were completely smooth and therefore void of channels), the deepest point of the draft would be shallower than if the melt were concentrated in basal channels.

Following this argument, we believe it is feasible that the presence of channels and channelised melting could lead to the persistence of pinning points and promote re-grounding. However, we have not made our perspective clear in the manuscript. We have added the following sentence, which is now on lines 257-258, to clarify what we mean: 'This ice base represents what we might expect if channels and channelised melt didn't exist and hence if all ice shelf melt were uniformly distributed across the ice shelf.'

Lines 268-269: "In the area shown, basal channels are clearly present within the thickness map (Figure 5a, marked by the cyan and magenta arrows). However they cannot be seen in the thickness change variables (Figures 5b-d)." Maybe it's worth mentioning here that this isn't necessarily very surprising, and it doesn't indicate that there's something wrong with the measurements. The calculations are Lagrangian, so all the lack of channelized pattern in the dH/dt and flux divergence says is that the parcel in the downstream measurement hasn't changed a whole lot as compared to when it was at the

upstream measurement, and that change is similar inside channels as outside channels. Or in this case, it just means that the amplitude of the channel isn't changing very much as you move downstream.

Thanks for addressing this. We have added the following to lines 276-278: 'This isn't necessarily surprising, it simply suggests that in this location the change in Lagrangian thickness and flux divergence is similar inside and out of the channel.'

Line 313: "Channel's" should be "channels," and near the end of the line, "channels" should be "channel's"

Thanks for pointing these out – we have now changed them.

Line 315: "depth-dependent" should be hyphenated

Thanks for pointing this out - it has been changed.

Line 321: "Smaller channels, approximately 1–1.5 km wide, from the central, narrow feature." This isn't a complete sentence.

Thanks for picking this up. This sentence has been changed to: 'Smaller channels, approximately 1-1.5 km wide, branch off the central feature', on like 329.

Line 323: "near" should probably be "nearly"

**This has been changed.**

Line 337: "across-channel" should be hyphenated

**This has been changed.**

Line 396: "it's" should be "its"

**This has been changed.**

Lines 398-399: "we believe these uncertainties do not alter our conclusions" probably needs to be backed up. Why do you believe this? Do you believe they're consistent in time? Or too small compared to the signal?

Thanks for highlighting the need to justify this statement. It is indeed because the errors associated with CryoSat-2 and ITS\_LIVE data are consistent in time. We have changed the end of this paragraph on lines 406-408 to say: 'Although we have not completed a formal error propagation here, we believe these uncertainties do not alter our conclusions as they are likely consistent in time. However, a full investigation into how different datasets used impact ice shelf basal melt rate estimation would be of benefit to the community.'

Line 431: "across-channel" should be hyphenated

**This has been changed.**

Lines 445-447: "Despite this asymmetry in melt rates, the channel apex doesn't deviate from the flow lines when temporal variability of ice velocity is taken into account (Figure 7i). This is contrary to suggestions from previous observations (Alley et al., 2024) and some modelling studies (Sergienko, 2013)..." Be very careful with this. The results from Alley et al. (2024) and Sergienko (2013) suggest that basal channels should deviate somewhat from flow lines in steady state – i.e., if you look at a plan-view image of a basal channel, and you look at plan-view flow lines that take into account the paths that ice parcels have taken throughout their journey across the ice flow (i.e. including history of ice-flow changes), the channel will be deflected slightly to the left of those flow lines when looking downstream.

This isn't an analysis that you've done in this paper. The two Eulerian analyses in Figure 7 would only show leftward migration if you have a non-steady Coriolis-influenced melt signal, i.e. if the Coriolis-influenced melt at that location is increasing over time. If your channels are not melting more intensely over time, which appears to be consistent with your data, you would not expect to see leftward migration in these. Your Lagrangian analysis in Figure 7 should show leftward migration, but only where you have Coriolis-influenced melt on the left-hand side of the channel. Figure 8 suggests that this is only true near the head of the channel, and Channel 1 certainly shows somewhat of a curve in that direction in the upper reaches. If your melt rates are correct, this almost has to be true – if it's melting faster on that side, the apex of the channel will necessarily move in that direction as the ice advects downstream. Your meltrate diierences between flanks decrease or reverse downstream, so we'd expect to see that eiect disappear or reverse (and I'm guessing it will be very small either way with those similar magnitudes downstream). Furthermore, the smaller branches you show downstream "veer westward" in the Lagrangian analysis, which would be consistent with Coriolis-favored melt in these features. I think this paragraph needs to be reworded, because in this context I don't think any of the papers cited and the analysis presented here are inconsistent with each other.

We would like to thank the reviewer for the detail with which they have thought about the above analysis. Having gone back through the manuscript, we certainly agree with many of their points.

Regarding the Eulerian analysis in Figure 7, the reviewer is correct that we would not expect to see channel deviation unless the Coriolis-favoured melting increases in time. We do not see this within our data, and hence the channels show no westward deviation. Under careful inspection, Channel 1 does deviate westward within the first year of observations in the Lagrangian analysis of Figure 7. However, it looks like this is more the channel getting wider as opposed to the apex deviating.

However, the analysis in Figure 9 (which is closer to the grounding line and hence where we see a bigger across-channel variation in melt) shows that the apex of Channel 1 does initially migrate west. The rate of that migration slows after the 2016 observations (consistent with the results presented in Figure 8).

These two pieces of evidence suggest that it is entirely possible for a channel apex to deviate from flow or for a channel to widen as a result of Coriolis-favoured melting. Which of these materialises likely depends on the amplitude of melt, the difference in melt across the channel and the flux divergence of the ice.

As a result of the reviewers comments, we have changed this paragraph to say:

'Consistent with previous observations (Alley et al., 2024) and modelling studies (Sergienko, 2013), the asymmetry in melt rates across Channel 1 near the grounding line leads to the westward deviation of the channel's apex from flow (Figure 9) and the widening of the channel on the western side (Figure 7i). Despite the reversal of across-channel melting downstream, we don't observe a recentring of the channel apex. This is likely because the melt signal is much smaller downstream of the grounding line. The rate of deviation from flow likely depends on the melt amplitude, the across-channel variation in melt and the flux divergence of the ice. We emphasise that these are merely observations from a single channel and further observations across a number of different channels on different ice shelves are required to gain a more in-depth understanding of this relationship. '

---

## Author Response (AR3)

**Responses to Editor Comment**

Comments from the reviewer are given in black.

Author responses are in red, and amendments to the manuscript are given in bold red.

We firstly want to thank the Editor for both their contribution to the reviews and for managing the whole submission process. Their expertise have been very helpful in improving and finalising the manuscript.

Figure 1: A' is not visible. Consider adding a semi-transparent box. The same holds for Figure 4b.

Thanks for pointing this out! We have slightly move the location of the A' marker and made them bold. The updated version can be seen in Figure E1. We have also made the transects and labels white in the Figure 4b, these can be seen in Figure E2.

Figure 6d: The labels "thickness" and "melt" are misleading because it is the anomaly that is shown (thickness cannot be negative). Change labels to include "anomaly".

Thanks for highlighting this. We have updated the figure to have 'anomaly' after each of the variable labels. The updated version can be seen in Figure E3.

l 144: The RACMO SMB is > 5 km grid and hence undersamples ice-shelf channels. There is some evidence that the surface mass balance is asymmetric across the channels and this can contribute to ice-shelf channel migration away from steady-state flowlines (https://doi.org/10.1029/2020JF005587). In your case this is likely a minor effect because the inferred ocean induced melt rates are much larger than the surface accumulation rates. Nevertheless, it could add to the differential signal that you see in Figure 7. Feel free to include this reference or to this disregard it, both is fine with me b/c it is a comparatively minor add-on.

Thanks for this suggestion. We think the suggested point is of value to the paper and has been included. On lines 190-193, we have added: 'We also note that the RACMO SMB data have relatively low spatial resolution compared to the width of channels, leading to undersampling across the channels. Although some evidence suggests that SMB can vary asymmetrically across Channels (Drews et al., 2020), this is unlikely to affect our results, as the basal melt signal is much larger than the SMB signal.'

l 268: A possibly relevant reference for density variability across ice-shelf channels is https://doi.org/10.5194/tc-10-811-2016 . We found some evidence that firn density may be elevated in ice-shelf channels because of ice layers which form as a consequence of by surface melt water collecting in the surface depressions. Feel free to include this reference or to this disregard it, both is fine with me b/c it is a comparatively minor add-on.

Thanks for the suggestion here, we also think this point is of value to the manuscript. Lines 268-271 now read: 'Channelised structures are also under-represented on the ice surface due to bridging stresses (Wearing et al., 2021; Drews, 2015) and kilometre-scale gradients in ice density (Dutrieux et al., 2013) potentially driven by surface melt-water pooling in channels and increasing the firn density (Drews et al., 2016).'

l 463: In view of the re-review this sentence appears to be part of an older version:

"Concentrating melt within channels promotes the formation of deeper keels compared to scenarios with spatially uniform melting, thereby increasing the likelihood of grounding."

I agree with the reviewer who mentioned that this type of framing is misleading. Please rephrase as suggested in your response, so that it becomes more clear that you refer to an otherwise spatially uniform melt rate that would make the ice shelf overall thinner (independent of ice-shelf channel keels).

Thanks for pointing this out. We had update the phrasing of this point in the Results section but missed it here in the Discussion. We have now updated this to say: 'If the ice shelf was to experience spatially uniform melting, the deepest part of the ice base would be shallower than the keel associated with channelised melt. Hence, a channel-keel geometry increases the likelihood of grounding.'

Data availability needs to be stronger in the next step. Make sure that DOIs are available, otherwise this may delay final publication.

The DOIs are being created by the Polar Data Centre. We expect them to be ready in 2 weeks. **Once available, we will include them in the manuscript.**

Figure E1: Updated version of Figure 1 with bold labels in (a).

Figure E3: Updated version of Figure 6 with 'anomaly' added to each of the variable labels.